# Catalytic activity and autoprocessing of murine caspase-11 mediate noncanonical inflammasome assembly in response to cytosolic LPS

Daniel C Akuma[1†], Kimberly A Wodzanowski[2†], Ronit Schwartz Wertman[1†], Patrick M Exconde[3], Víctor R Vázquez Marrero[2], Chukwuma E Odunze[4], Daniel Grubaugh[1], Sunny Shin[2*], Cornelius Taabazuing[3*], Igor E Brodsky[1*]

[1]Department of Pathobiology, University of Pennsylvania School of Veterinary Medicine, Philadelphia, United States; [2]Department of Microbiology, University of Pennsylvania Perelman School of Medicine, Philadelphia, United States; [3]Department of Biochemistry and Biophysics, University of Pennsylvania Perelman School of Medicine, Philadelphia, United States; [4]University of Maryland, College Park, College Park, United States

*For correspondence:
sunshin@pennmedicine.upenn.
edu (SS);
Cornelius.Taabazuing@
Pennmedicine.upenn.edu (CT);
ibrodsky@vet.upenn.edu (IEB)

[†]These authors contributed
equally to this work

Reviewing Editor: Jungsan
Sohn, Johns Hopkins University
School of Medicine, United
States

**Abstract** Inflammatory caspases are cysteine protease zymogens whose activation following infection or cellular damage occurs within supramolecular organizing centers (SMOCs) known as inflammasomes. Inflammasomes recruit caspases to undergo proximity-induced autoprocessing into an enzymatically active form that cleaves downstream targets. Binding of bacterial LPS to its cytosolic sensor, caspase-11 (Casp11), promotes Casp11 aggregation within a high-molecular-weight complex known as the noncanonical inflammasome, where it is activated to cleave gasdermin D and induce pyroptosis. However, the cellular correlates of Casp11 oligomerization and whether Casp11 forms an LPS-induced SMOC within cells remain unknown. Expression of fluorescently labeled Casp11 in macrophages revealed that cytosolic LPS induced Casp11 speck formation. Unexpectedly, catalytic activity and autoprocessing were required for Casp11 to form LPS-induced specks in macrophages. Furthermore, both catalytic activity and autoprocessing were required for Casp11 speck formation in an ectopic expression system, and processing of Casp11 via ectopically expressed TEV protease was sufficient to induce Casp11 speck formation. These data reveal a previously undescribed role for Casp11 catalytic activity and autoprocessing in noncanonical inflammasome assembly, and shed new light on the molecular requirements for noncanonical inflammasome assembly in response to cytosolic LPS.

## Editor's evaluation

This fundamental work advances our understanding of how caspase-11 is regulated by LPS. The evidence supporting the conclusions is compelling, with rigorous biochemical and cellular studies. The work will be of broad interest to immunologists and biochemists.

## Introduction

The mammalian innate immune system relies on evolutionarily conserved pattern recognition receptors (PRRs) to detect pathogen-associated molecular patterns (PAMPs) in order to rapidly respond to potential microbial threats (*Blander and Sander, 2012*; *Janeway, 1989*; *Janeway and Medzhitov,*

*2002*). Such swift responses are critical for efficient host defense in the face of pathogens that have the capacity to replicate rapidly and suppress or evade immune detection. A common feature of PRR activation is their inducible assembly into higher-order oligomeric protein complexes, termed supramolecular organizing centers (SMOCs), which mediate the effector function of the PRR signaling pathway in response to microbial infection or pathologic stimulus (*Kagan et al., 2014*).

Caspases-1 and -11 are inflammatory caspases that are recruited into inflammasome SMOCs in response to the cytosolic presence of pathogen-derived signals or molecules (*Broz and Dixit, 2016*; *Lamkanfi and Dixit, 2014*). Inflammatory caspases are zymogens that contain a C-terminal enzymatic domain and an N-terminal caspase activation and recruitment domain (CARD), which promotes SMOC formation through homotypic interactions with other CARD-containing adapter proteins as well as homodimerization with each other (*Lu et al., 2014*; *Park et al., 2007*; *Ting and Davis, 2005*). The C-terminal enzymatic domain of caspases is comprised of large (17–20 kDa) and small (10–12 kDa) subunits separated by an interdomain linker (IDL), which contains the self-cleavage site that undergoes autoprocessing following caspase oligomerization (*Ross et al., 2022*; *Walker et al., 1994*; *Wilson et al., 1994*).

Caspase-1 is recruited to canonical inflammasome complexes that typically contain a sensor protein of the NLR family, the adaptor protein ASC, and caspase-1 itself (*Broz and Dixit, 2016*; *Lamkanfi and Dixit, 2014*). In contrast, caspase-11 (Casp11) binds directly to lipopolysaccharide (LPS) from Gram-negative bacteria (*Shi et al., 2014*) and does not require a known sensor or adapter for activation of pyroptosis via a pathway termed the noncanonical inflammasome (*Kayagaki et al., 2011*). Recruitment of Casp1 and Casp11 to their respective SMOCs results in their autoprocessing, a process termed proximity-induced activation, enabling their cleavage of downstream targets (*Salvesen and Dixit, 1999*; *Shi, 2004*). Casp11-dependent cleavage of its target, gasdermin D (GSDMD), leads to cell lysis, as well as release of inflammatory cellular contents, thus triggering pyroptosis, a pro-inflammatory programmed cell death (*Aglietti et al., 2016*; *Kayagaki et al., 2015*; *Shi et al., 2015*). As a result, Casp1 and Casp11 play important roles in antimicrobial host defense, but they can also promote autoinflammatory disease when dysregulated (*Franchi et al., 2009*; *Hagar et al., 2013*; *Henao-Mejia et al., 2012*; *Kayagaki et al., 2013*).

Inducible dimerization is used to model assembly of functional caspase complexes within cells (*Ball et al., 2020*; *Boucher et al., 2018*; *Oberst et al., 2010*; *Ross et al., 2018*). Caspases whose N-terminal domain is replaced with a dimerizable domain, such as FKBP506, undergo autoprocessing and exhibit catalytic activity toward their substrates in the presence of dimerizing agents. Interestingly, Casp11 mutants, whose autoprocessing site is ablated, have significantly reduced protease activity toward their substrates and have defective ability to induce cell death, despite being dimerized and having an intact catalytic site (*Ross et al., 2018*). Furthermore, CRISPR targeting of Casp11 autoprocessing or catalytic activity demonstrates that both are important for Casp11-dependent pyroptosis and lethal sepsis (*Lee et al., 2018*).

The observation that autoprocessing is needed to generate a fully functional Casp11 complex despite being dimerized and catalytically active (*Ross et al., 2018*) implies that formation of a functional Casp11 SMOC involves additional steps beyond inducible dimerization. Interestingly, while non-cleavable Casp8 is active in vitro (*Chang et al., 2003*) or in the presence of kosmotropic salts (*Pop et al., 2007*), autoprocessing is also required for full Casp8 activity toward its substrates within cells, implying that autoprocessing stabilizes the active form of the enzyme (*Oberst et al., 2010*). This parallels recent findings with Casp1, which requires autoprocessing for robust cleavage of IL-1β (*Broz et al., 2010*) and GSDMD (*Ball et al., 2020*). However, while ASC fluorescent reporters have enabled the dynamic tracking of canonical inflammasome assembly within cells (*Stutz et al., 2013*; *Tzeng et al., 2016*), the cellular correlates of Casp11 oligomerization within cells in response to cytosolic LPS remain poorly defined.

Here, we find using fluorescent Casp11 reporter fusions that cytosolic LPS induces Casp11 assembly into large perinuclear specks, the first time, to our knowledge, that LPS-induced noncanonical inflammasome speck assembly has been directly observed within cells. Unexpectedly, Casp11 catalytic activity and autoprocessing were required for cytosolic LPS-inducible Casp11 speck formation in macrophages, suggesting that both catalytic activity and autoprocessing act *upstream* of Casp11 oligomerization to facilitate assembly of the fully activated Casp11 SMOC in response to cytosolic LPS. Intriguingly, both Casp11 catalytic activity and autoprocessing were required for spontaneous

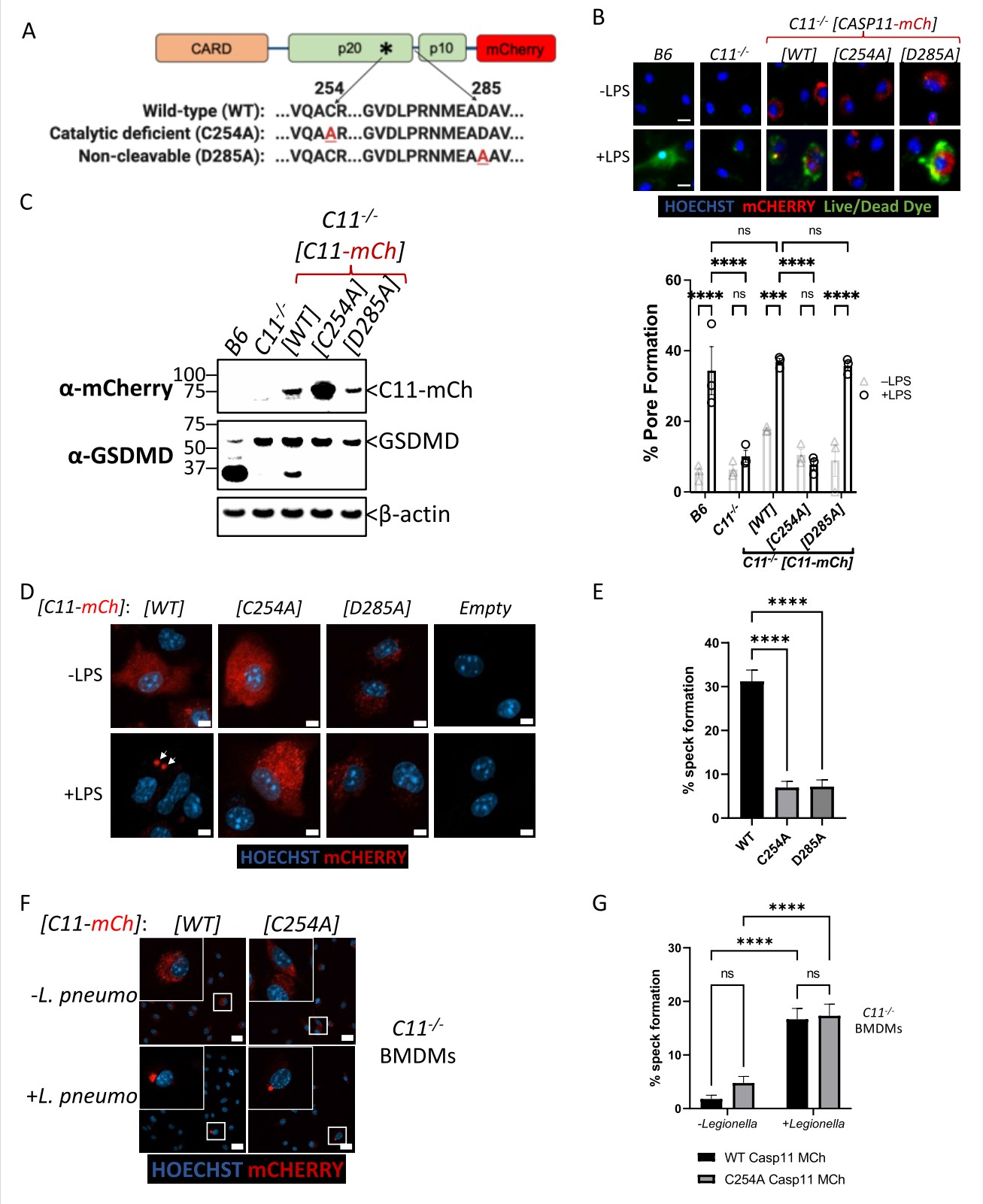

**Figure 1.** Caspase-11 catalytic activity and autoprocessing are required for cytosolic lipopolysaccharide (LPS)-induced speck formation. (**A**) Schematic representation of Casp11 fluorescent reporter constructs used in this study indicating mCherry fused to the C-terminus of wild-type (WT), catalytically inactive (C254A), or non-cleavable (D285A) Casp11. (**B–E**) C57BL/6 (B6) or *Casp11⁻ᐟ⁻* primary bone marrow-derived macrophages (BMDMs) with the indicated transgenes or empty-vector control were primed with Pam3CSK4 for 4 hr, followed by transfection with LPS (2 μg/mL) from *S. enterica* serovar

*Figure 1 continued on next page*

*Figure 1 continued*

Minnesota. (**B**) Caspase-11-mediated pore formation was assayed and quantified 8 hr later as percentage of B6 cells or Casp11-mCherry-expressing cells that took up the Live/Dead Green Fluorescent dead cell dye (Invitrogen). Nuclei are stained with Hoechst. (**C**) Casp11-mCherry expression and gasdermin D (GSDMD) processing in response to LPS (2 μg/mL) transfection were assessed (8 hr post-transfection) by western blotting for mCherry and GSDMD as indicated. β-actin was used as a loading control. (**D, F**) Cells expressing indicated Casp11-mCherry constructs were fixed 6 hr post-LPS transfection (**D**) or *Legionella pneumophila* infection (**F**, MOI (multiplicity of infection) of 50) and prepared for confocal microscopy. Nuclei are stained with Hoechst. White arrows indicate Casp11-mCherry specks. Scale bar, 10 μm. (**E, G**) Speck formation from cells in (**D**) and (**F**), respectively, was quantified as percentage of Casp11-mCherry-expressing cells containing a speck. Each data point represents four image frames (100–150 cells) per well and three wells per condition for a total of 300–450 cells. All error bars represent mean ± SEM of triplicate wells; representative of three independent experiments. ***p<0.001, ****p<0.0001, ns, not significant. Two-way ANOVA with Sidak's multiple-comparison test.

The online version of this article includes the following source data for figure 1:

**Source data 1.** Source data for *Figure 1B*.

**Source data 2.** Source data for *Figure 1C*.

**Source data 3.** Source data for *Figure 1D*.

**Source data 4.** Source data for *Figure 1F*.

as well as LPS-induced speck assembly in HEK293T cells, consistent with a model whereby higher-order oligomerization of Casp11 requires catalytic activity and autoprocessing. Consistently, inducible IDL processing of Casp11 by an exogenous protease was sufficient to mediate speck formation in 293T cells. Surprisingly, the Casp11 CARD was neither required nor sufficient for spontaneous Casp11 oligomerization, as inducible dimerization of Casp11 by a chemical dimerizer in the absence of the CARD promoted speck formation, and a CARD-less Casp11 p20/p10-mCherry construct also exhibited speck formation in a manner that also required catalytic activity. Altogether, we demonstrate that Casp11 catalytic activity and autoprocessing act upstream of Casp11 oligomerization to mediate assembly of the fully functional Casp11 SMOC in response to cytosolic LPS. Our findings uncover a previously undescribed property of caspase-containing SMOCs and imply that Casp11 activation involves a feed-forward mechanism whereby self-processing of Casp11 facilitates assembly of the fully competent pyroptosis-inducing Casp11 complex.

## Results

### Caspase-11 catalytic activity is required for cytosolic LPS-induced speck formation

Upon sensing cytosolic LPS, Casp11 oligomerizes into high-molecular-weight complexes known as noncanonical inflammasomes (*Shi et al., 2014*). To better understand the oligomerization dynamics of the Casp11 inflammasome, we generated a lentiviral construct encoding C-terminal fusions of mCherry with full-length Casp11 (Casp11$^{WT}$-mCherry), along with corresponding mutant constructs ablating catalytic activity (Casp11$^{C254A}$-mCherry), or the IDL autoprocessing site (Casp11$^{D285A}$-mCherry), and transduced these constructs into *Casp11$^{-/-}$* bone marrow-derived macrophages (BMDMs) (*Figure 1A*). Casp11$^{WT}$-mCherry, but not Casp11$^{C254A}$-mCherry, restored GSDMD cleavage and pore formation in *Casp11$^{-/-}$* BMDMs, indicating that the tagged Casp11-mCherry construct retains functionality (*Figure 1B, C*). Interestingly, Casp11$^{D285A}$-mCherry expression failed to completely ablate pore formation despite abrogating GSDMD cleavage, indicating that tagged Casp11-mCherry carries out GSDMD cleavage in a manner dependent on its autoprocessing (*Figure 1B, C*). We next utilized confocal microscopy to assess Casp11-mCherry localization at basal state and in response to cytosolic LPS. As expected, *Casp11$^{-/-}$* BMDMs transduced with Casp11$^{WT}$-mCherry, Casp11$^{C254A}$-mCherry, or Casp11$^{D285A}$-mCherry exhibited diffuse cytoplasmic staining in the absence of LPS stimulation (*Figure 1D*). Notably, Casp11$^{WT}$-mCherry coalesced into large specks in response to transfection of LPS, consistent with the concept that Casp11 assembles into an LPS-induced higher-order complex, or SMOC (*Figure 1D, E*). In contrast, Casp11$^{C254A}$-mCherry and Casp11$^{D285A}$-mCherry-transduced cells were unable to form specks in response to LPS (*Figure 1D, E*). These findings imply that CARD-dependent binding of Casp11 to LPS is not sufficient to mediate Casp11 speck formation within cells, and that Casp11 catalytic activity and autoprocessing facilitate higher order assembly of noncanonical inflammasomes. As these experiments utilized delivery of purified LPS into the cytosol, we

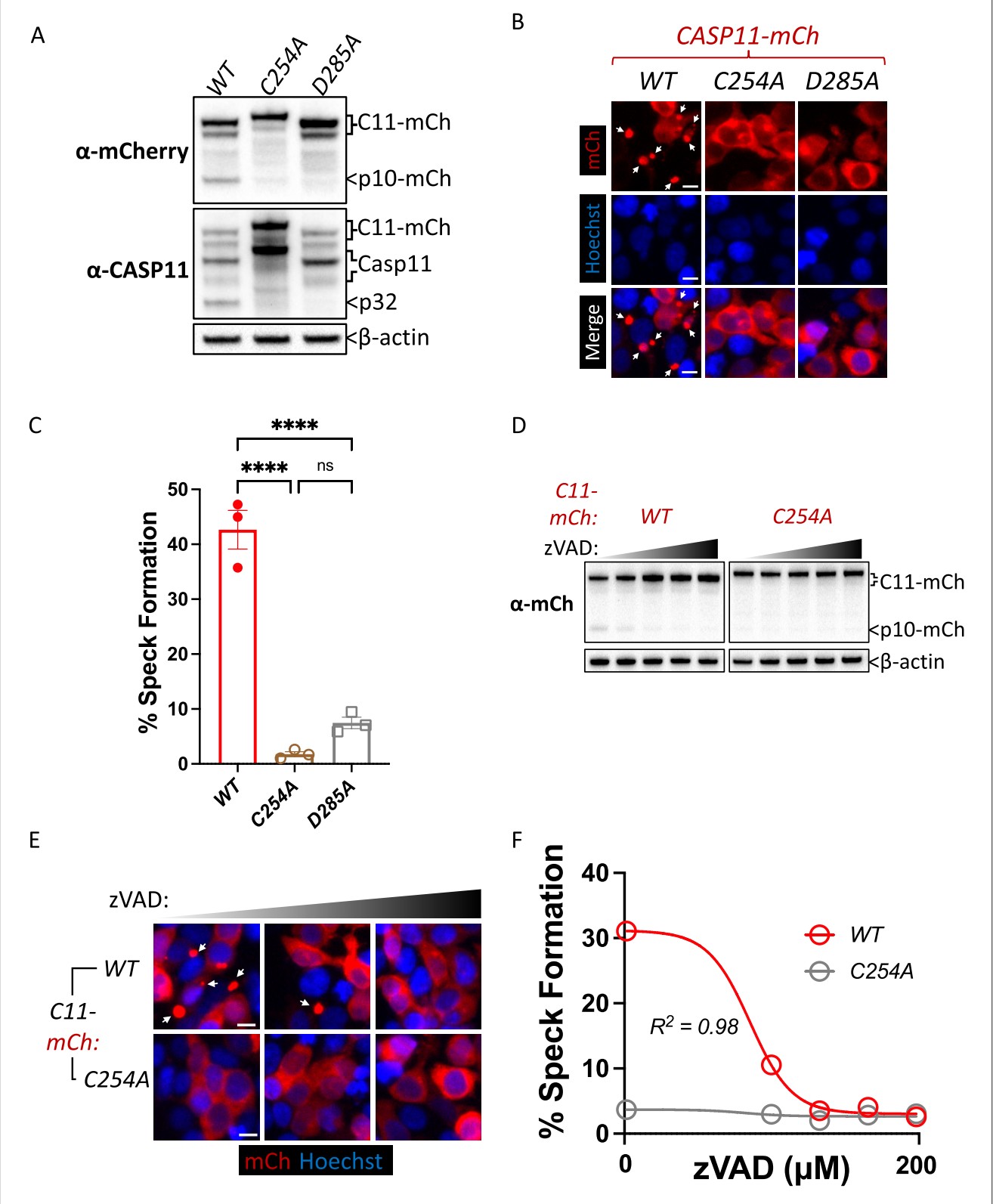

**Figure 2.** Caspase-11 catalytic activity and autoprocessing at the interdomain linker are required for spontaneous caspase-11 oligomerization in HEK293T cells. (**A**) Wild-type (WT), catalytically inactive (C254A), and non-cleavable (D285A) Casp11-mCherry expression plasmids were transfected (0.25 μg) into HEK293T cells. Cell lysates were immunoblotted for mCherry, Casp11, and β-actin (loading control) 10 hr post-transfection. (**B**) HEK293T cells were transfected with wild-type (WT), catalytically inactive (C254A), or non-cleavable (D285A) Casp11-mCherry as described in 'Materials and

*Figure 2 continued on next page*

*Figure 2 continued*

methods' and imaged by fluorescence microscopy 18 hr post-transfection. Nuclei (blue) were stained with Hoechst, white arrows denote Casp11-mCherry specks. Scale bar = 10 μm. (**C**) Speck formation in (**B**) was quantified as percentage of Casp11-mCherry-expressing cells containing at least one speck. (**D**) HEK293T cells were transfected with Casp11-mCherry constructs as in (**B**) and 6 hr post-transfection, and cells were incubated with increasing amounts of pan-caspase inhibitor zVAD (0–200 μM; twofold increments). Whole-cell lysates were isolated 12 hr post-transfection and immunoblotted for mCherry or β-actin loading control as indicated. Cleaved p10-mCherry is denoted. (**E**) Casp11-mCherry speck formation was assayed in zVAD-treated cells by fluorescence microscopy as in (**B**). (**F**) Speck formation in (**E**) was quantified as percentage of Casp11-mCherry-expressing cells containing at least one speck. Dose–response curves in (**F**) were plotted by least-squares nonlinear regression ([Log$_2$(inhibitor) vs. response (three parameters)]; Y = Bottom + (Top-Bottom)/(1 + 10$^{(X-LogIC50)}$); R$^2$ indicated). Error bars represent mean ± SEM of triplicate wells (800–900 cells per well); representative of 2–3 independent experiments. Bar graphs in (**C**) were analyzed by two-way ANOVA with Sidak's multiple-comparison test, ***p<0.001.

The online version of this article includes the following source data and figure supplement(s) for figure 2:

**Source data 1.** Source data for *Figure 2A*.

**Source data 2.** Source data for *Figure 2B*.

**Source data 3.** Source data for *Figure 2D*.

**Source data 4.** Source data for *Figure 2E*.

**Source data 5.** Source data for *Figure 2F*.

**Figure supplement 1.** Casp11-mCherry maintains enzymatic function in HEK293T cells.

**Figure supplement 1—source data 1.** Source data for *Figure 2—figure supplement 1A*.

**Figure supplement 2.** Casp11 catalytic activity mediates gasdermin D (GSDMD) cleavage, pyroptosis, and IL-1β release in response to intracellular lipopolysaccharide (LPS) in primary bone marrow-derived macrophages (BMDMs).

**Figure supplement 2—source data 1.** Source data for *Figure 2—figure supplement 2C*.

**Figure supplement 3.** Catalytic activity is not required for caspase-11 intermolecular interactions.

**Figure supplement 3—source data 1.** Source data for *Figure 2—figure supplement 3B*.

infected Casp11$^{WT}$-mCherry- and Casp11$^{C254A}$-mCherry-expresssing BMDMs with an intracellular Gram-negative pathogen, *Legionella pneumophila*, which is known to activate the noncanonical inflammasome (*Case et al., 2013*; *Casson et al., 2013*), in order to test the ability of Casp11 to form specks in the context of bacterial infection. Notably, cells expressing Casp11$^{WT}$-mCherry exhibited speck formation upon infection with *L. pneumophila*, similarly to cells exposed to transfected LPS alone (*Figure 1F, G*). Interestingly, *L. pneumophila* infection also induced speck formation in cells expressing Casp11$^{C254A}$-mCherry, in contrast to our findings with cytosolic LPS (*Figure 1F, G*). These data suggest that in addition to LPS itself, other properties associated with intact bacteria promote Casp11 oligomerization and inflammasome activation.

## Caspase-11 catalytic activity and autoprocessing at the interdomain linker are required for spontaneous caspase-11 oligomerization in HEK293T cells

Our observation that catalytic activity and autoprocessing were essential for cytosolic LPS-induced aggregation in macrophages suggested that Casp11 catalytic activity contributes to higher-order complex assembly, rather than higher-order complex formation being due solely to Casp11 binding to LPS. To further dissect these mechanisms, we employed an extensively-used ectopic expression system to define the molecular basis for inflammasome assembly in HEK293T cells, which normally do not express inflammasome components (*Kayagaki et al., 2015*; *Rauch et al., 2017*; *Ross et al., 2018*; *Shi et al., 2014*; *Tenthorey et al., 2014*). In 293T cells, ectopic Casp11 expression mediates activation of the noncanonical inflammasome and processing of GSDMD (*Aglietti et al., 2016*; *Kayagaki et al., 2015*; *Shi et al., 2015*). Consistent with previous observations, co-expression of Casp11$^{WT}$-mCherry and GSDMD led to dose-dependent GSDMD cleavage and cell death that required Casp11 catalytic activity and autoprocessing (*Figure 2A*, *Figure 2—figure supplement 1*; *Lee et al., 2018*; *Ross et al., 2018*). Notably, Casp11$^{WT}$-mCherry spontaneously assembled into large aggregates reminiscent of ASC-containing specks (*Stutz et al., 2013*; *Tzeng et al., 2016*), whereas Casp11$^{C254A}$-mCherry and Casp11$^{D285A}$-mCherry remained diffuse throughout the cytosol, consistent with our previous finding in macrophages that Casp11 catalytic activity and autoprocessing are necessary to initiate or propagate Casp11 inflammasome assembly (*Figure 2B and C*). Moreover, the pan-caspase inhibitor Z-VAD-FMK

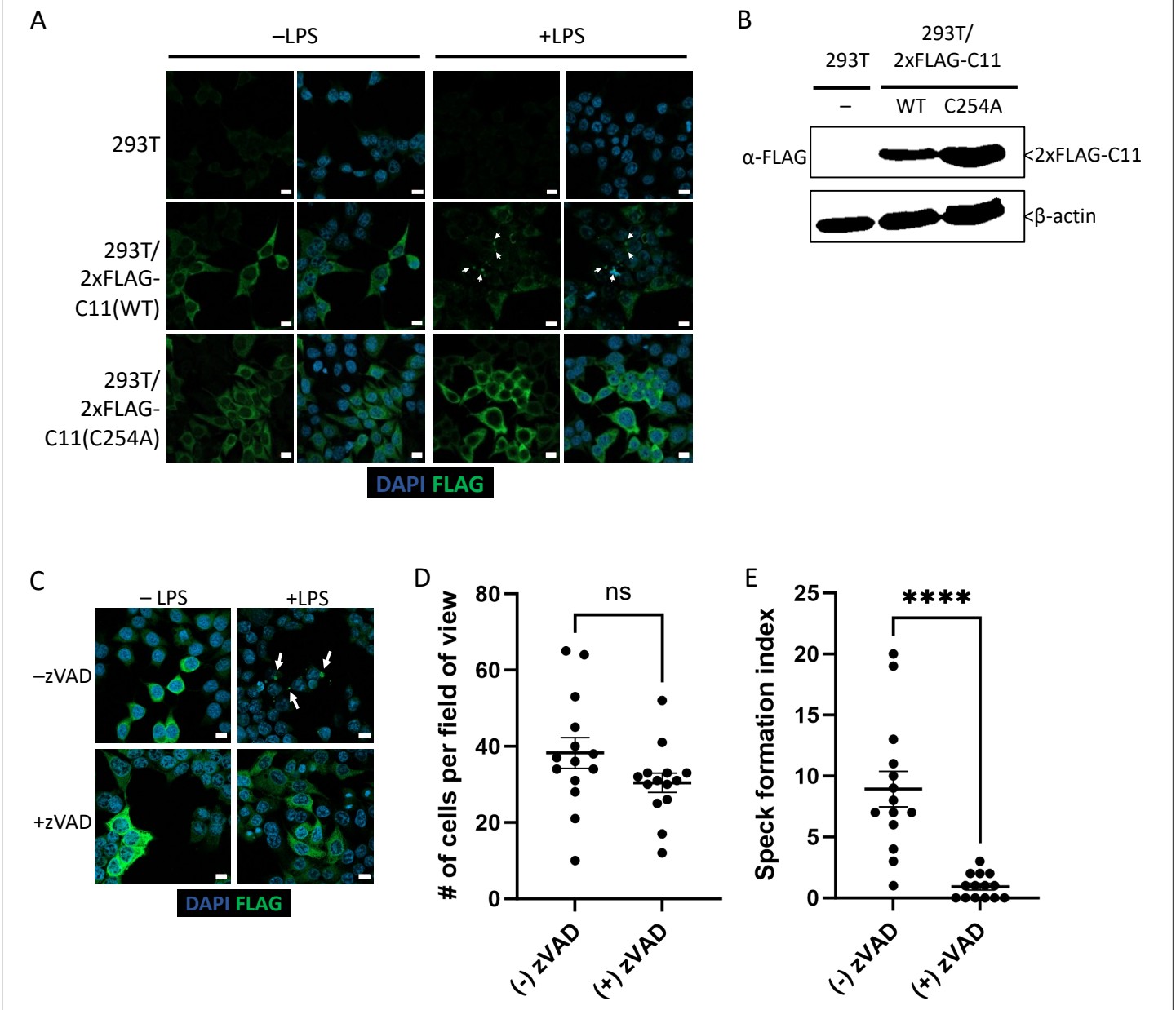

**Figure 3.** Caspase-11 catalytic activity is required for lipopolysaccharide (LPS)-induced oligomerization in HEK293T cells. (**A**) HEK293T cells stably expressing 2xFLAG-Casp11 (WT) or 2xFLAG-Casp11 (C254A) were transfected with LPS (1 µg/mL) from *S. enterica* serovar Minnesota and imaged by fluorescence microscopy 24 hr post-transfection. Casp11 was stained with anti-FLAG-FITC (green), and nuclei (blue) were stained with DAPI. White arrows denote Casp11 specks. Scale bar = 10 µm. (**B**) Casp11 protein levels in each stable HEK293T cell line were assayed by immunoblotting for FLAG or β-actin (loading control). (**C**) HEK293T cells stably expressing WT 2xFLAG-Casp11 were transfected with LPS as in (**A**), in the presence of pan-caspase inhibitor zVAD (200 µM). Casp11 was stained by immunofluorescence using anti-FLAG-FITC (green) and imaged by confocal microscopy (×63 objective). Nuclei (blue) were stained with DAPI. White arrows denote Casp11 specks. Scale bar = 10 µm. Speck formation in (**C**) was quantified by (**D**) number of cells per field of view, and (**E**) number of cells per field of view with at least one speck, denoted as 'speck formation index.' Error bars represent mean ± SEM of 14 fields of view (500 cells) from three independent experiments. Data were analyzed by Student's *t*-test, ns, not significant, ****p<0.0001.

The online version of this article includes the following source data and figure supplement(s) for figure 3:

**Source data 1.** Source data for *Figure 3B*.

**Figure supplement 1.** N- and C-terminal caspase-11 subunits co-localize in caspase-11 specks.

**Figure supplement 1—source data 1.** Source data for *Figure 3—figure supplement 1B*.

(zVAD) blocked spontaneous speck assembly and autoprocessing by Casp11$^{WT}$-mCherry in a dose-dependent manner (*Figure 2D–F*). Consistently, Casp11$^{C254A}$-mCherry did not aggregate into visible specks, and its cytoplasmic distribution was not detectably altered in the presence of zVAD. Similar doses of zVAD abrogated LPS-induced Casp11-mediated pyroptosis in BMDMs (*Figure 2—figure supplement 2A–C*). Importantly, and consistent with prior studies (*Shi et al., 2014*), catalytic activity was not required for Casp11 self-association per se, as Casp11$^{WT}$ and Casp11$^{C254A}$ associate both with themselves and each other in reciprocal co-immunoprecipitations (*Figure 2—figure supplement 3*). Together, these data imply that catalytic activity plays an important role in higher-order Casp11 oligomerization in response to cytosolic LPS.

## Caspase-11 catalytic activity is required for LPS-induced oligomerization in HEK293T cells

While transient transfection of Casp11$^{WT}$-mCherry led to spontaneous speck formation in HEK293T cells, we also generated stable Casp11$^{WT}$- and Casp11$^{C254A}$-expressing HEK293T cells in order to investigate whether stable expression of tagged Casp11 would lead to LPS-inducible specks in 293T cells. We also utilized an N-terminal 2xFLAG tag to address the potential impact of tag size and location on Casp11 speck assembly. In stably expressing cells, Casp11$^{WT}$ and Casp11$^{C254A}$ exhibited similar levels of expression and showed diffuse cytoplasmic staining (*Figure 3A, B*). Importantly, LPS transfection induced speck formation in 2xFLAG-Casp11$^{WT}$-expressing cells, but not in 2xFLAG-Casp11$^{C254A}$ cells, consistent with our findings in BMDMs (*Figure 3A*). Moreover, inhibition of catalytic activity with the pan-caspase inhibitor zVAD also blocked formation of specks in 2xFLAG-Casp11$^{WT}$-expressing cells, consistent with our previous findings (*Figure 3C–E*). Our finding that both N- and C-terminal portions of Casp11 were found within specks by microscopy, despite evidence that Casp11 processing was taking place, suggested that both the N- and C-terminal domains of Casp11 were simultaneously present within the same large specks. To directly test this hypothesis, we transiently transfected Casp11$^{WT}$-mCherry into HEK293T cells stably expressing 2xFLAG-Casp11$^{WT}$ or catalytically inactive 2xFLAG-Casp11$^{C254A}$. Notably, Casp11$^{WT}$-mCherry induced spontaneous formation of specks, which recruited WT 2xFLAG-Casp11, as expected (*Figure 3—figure supplement 1*). Critically, Casp11-mCherry specks also recruited catalytically inactive 2xFLAG-Casp11$^{C254A}$, demonstrating that both N- and C-termini are present within the specks and that catalytically inactive Casp11 can be recruited to complexes formed by the wild-type protein (*Figure 3—figure supplement 1*).

## Wild-type caspase-11 recruits catalytically inactive caspase-11 to speck complexes independently of trans-processing

Catalytically inactive Casp11 could be recruited to wild-type complexes via trans-processing of the inactive mutant by wild-type Casp11, or due to recruitment of unprocessed inactive Casp11 to nascent wild-type specks. To address this question, we employed transient co-transfection of wild-type and mutant versions of labeled and unlabeled Casp11, and initially co-transfected a fixed amount of mCherry-tagged Casp11$^{WT}$ or Casp11$^{C254A}$ with increasing amounts of unlabeled WT Casp11 (*Figure 4A*). Consistent with our previous observations, in the absence of unlabeled WT Casp11, Casp11$^{WT}$-mCherry exhibited autoprocessing and speck formation that was absent in Casp11$^{C254A}$-mCherry cells (*Figure 4B, left lanes, C*). Increasing levels of unlabeled Casp11 did not affect processing of Casp11$^{WT}$-mCherry, though it did increase processing of Casp11$^{C254A}$-mCherry (*Figure 4B*). Increasing levels of unlabeled Casp11 had a relatively limited effect on speck formation by Casp11$^{WT}$-mCherry, except at the very highest dose (*Figure 4C, D*). In contrast, and consistent with our observations in stable 2xFLAG-Casp11$^{C254A}$-expressing cells, co-transfection of unlabeled Casp11 significantly increased speck formation by Casp11$^{C254A}$-mCherry (*Figure 4C, D*). These findings demonstrate that WT Casp11 rescues oligomerization of catalytically inactive Casp11-mCherry species by enabling recruitment of the mutant protein to WT complexes. Intriguingly, mutating the IDL auto-processing site of Casp11$^{C254A}$-mCherry to yield an inactive, IDL-uncleavable mutant Casp11$^{C254A/D285A}$-mCherry prevented IDL processing but did not prevent formation of mCherry specks in the presence of unlabeled WT Casp11 (*Figure 4E–H*). Importantly, neither co-transfected Casp11$^{C254A}$-mCherry nor Casp11$^{C254A/D285A}$-mCherry impaired pyroptosis or GSDMD cleavage by WT Casp11 (*Figure 4—figure supplement 1*). Altogether, these studies demonstrate that catalytic activity and autoprocessing are

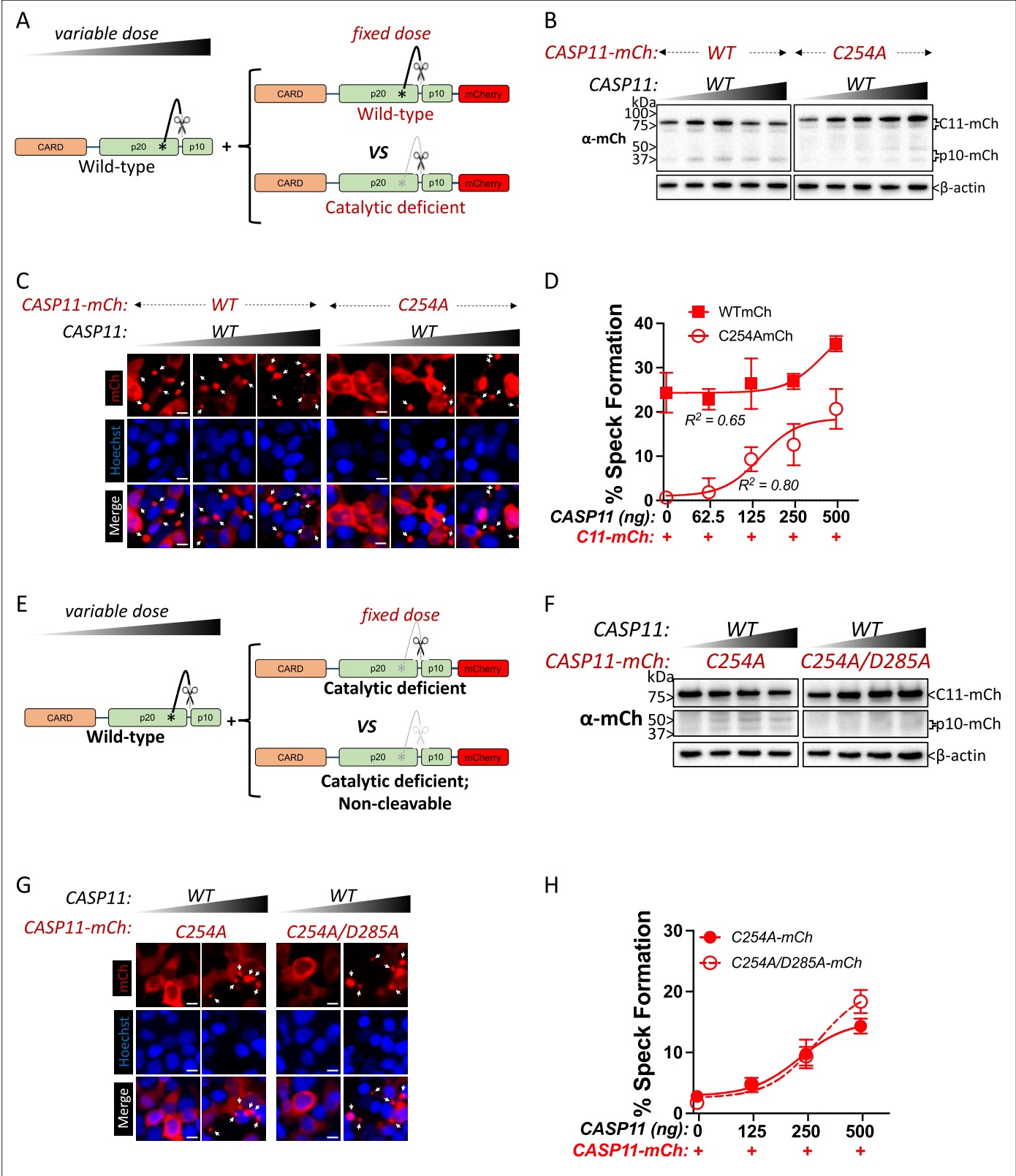

**Figure 4.** Wild-type caspase-11 recruits catalytically inactive caspase-11 to speck complexes independently of trans-processing. (**A, E**) Schematic diagram indicating co-transfection combinations used in (**B–H**). Untagged full-length wild-type (WT) caspase-11 gene constructs were transfected at increasing doses, together with a fixed amount of indicated mCherry-tagged Casp11. (**B**) 12 hr post-transfection, whole-cell lysates were harvested and immunoblotted for mCherry or β-actin as a loading control. (**C**) 18 hr following transfection, the cells were imaged by fluorescence microscopy. Nuclei

*Figure 4 continued on next page*

*Figure 4 continued*

(blue) are stained with Hoechst, white arrows represent Casp11 oligomers (specks), scale bar = 10 μm. (**D**) Speck formation was quantified as percentage of mCherry-expressing cells containing at least one speck. (**F**) Whole-cell lysates of indicated transfected cells was assayed as in (**B**). (**G**) Cells transfected with indicated constructs were imaged as in (**C**). (**H**) Speck formation with respect to increasing levels of WT Casp11 was plotted as in (**D**). Error bars indicate mean ± SEM of triplicate wells (800–900 cells per well); representative of 2–3 independent experiments. Dose–response curves were plotted by least-squares nonlinear regression ([Log$_2$(agonist) vs. response (three parameters)]; Y = Bottom + (Top-Bottom)/(1 + 10$^{(LogEC50-X)}$); R$^2$ indicated).

The online version of this article includes the following source data and figure supplement(s) for figure 4:

**Source data 1.** Source data for *Figure 4B*.

**Source data 2.** Source data for *Figure 4C*.

**Source data 3.** Source data for *Figure 4D*.

**Source data 4.** Source data for *Figure 4F*.

**Source data 5.** Source data for *Figure 4G*.

**Source data 6.** Source data for *Figure 4H*.

**Figure supplement 1.** Caspase-11 activity remains intact despite co-transfection with catalytically inactive caspase-11.

**Figure supplement 1—source data 1.** Source data for *Figure 4—figure supplement 1B*.

**Figure supplement 1—source data 2.** Source data for *Figure 4—figure supplement 1C*.

required for the formation of higher-order oligomeric Casp11 complexes, and that recruitment of inactive Casp11 to these complexes is independent of IDL trans-processing.

## Caspase-11 autoprocessing mediates noncanonical inflammasome assembly

The above observations imply that while catalytically deficient Casp11 lacks the ability to oligomerize autonomously, the presence of a wild-type partner can recruit catalytically inactive and autoprocesssing-deficient Casp11 to SMOCs formed by wild-type Casp11. To directly test whether catalytic activity and autoprocessing were both required within the same molecule to recruit catalytically deficient Casp11, we co-transfected tagged Casp11$^{C254A}$-mCherry with increasing levels of untagged Casp11$^{WT}$, catalytically inactive Casp11$^{C254A}$ or IDL-uncleavable Casp11$^{D285A}$ (*Figure 5A*). As expected, WT Casp11 could undergo autoprocessing, which required catalytic activity and IDL autoprocessing (*Figure 5B*). Consistent with our prior observations, WT Casp11 rescued speck formation by catalytically inactive Casp11$^{C254A}$-mCherry (*Figure 5C and D*). However, neither the catalytic Casp11$^{C254A}$ nor the autoprocessing Casp11$^{D285A}$ mutants could recruit Casp11$^{C254A}$-mCherry to specks (*Figure 5C and D*). Importantly, Casp11$^{C254A}$-mCherry did not impair Casp11-mediated cytotoxicity in HEK293T cells stably expressing GSDMD, as evidenced by LDH release (*Figure 5E*).

Altogether, these findings demonstrate a key role for Casp11 catalytic activity and autoprocessing in assembly of higher-order Casp11 inflammasome complexes. These findings also indicate that these activities must be present within the same molecule and occur *upstream of* Casp11 inflammasome assembly, rather than occurring downstream of inflammasome assembly.

## Catalytic activity is required for caspase-11 speck formation downstream of homodimerization

Homodimerization is thought to mediate activation of initiator caspases, as homodimerization leads to catalytic activity-dependent processing of caspase substrates (*Ball et al., 2020*; *Boucher et al., 2018*; *Oberst et al., 2010*; *Ross et al., 2018*). Our findings suggested that homodimerization might serve to nucleate inflammasome assembly, and that catalytic activity-dependent Casp11 oligomerization occurs subsequent to initial dimerization. To determine whether catalytic activity mediates higher-order Casp11 oligomerization downstream of initial dimerization, or whether speck formation proceeds downstream of homodimerization independently of catalytic activity, we replaced the endogenous CARD with a DmrB dimerization domain to generate dimerizable Casp11$^{WT}$- and Casp11$^{C254A}$-mCherry that could undergo inducible dimerization in the presence of a chemical dimerizer (AP20187) (*Figure 6A*). Interestingly, DmrB-(ΔCARD)-Casp11-mCherry displayed basal levels of speck formation above background, which was significantly increased by addition of dimerizer (*Figure 6B and C*). Critically, both baseline and dimerization-induced speck formation of DmrB-(ΔCARD)-Casp11-mCherry

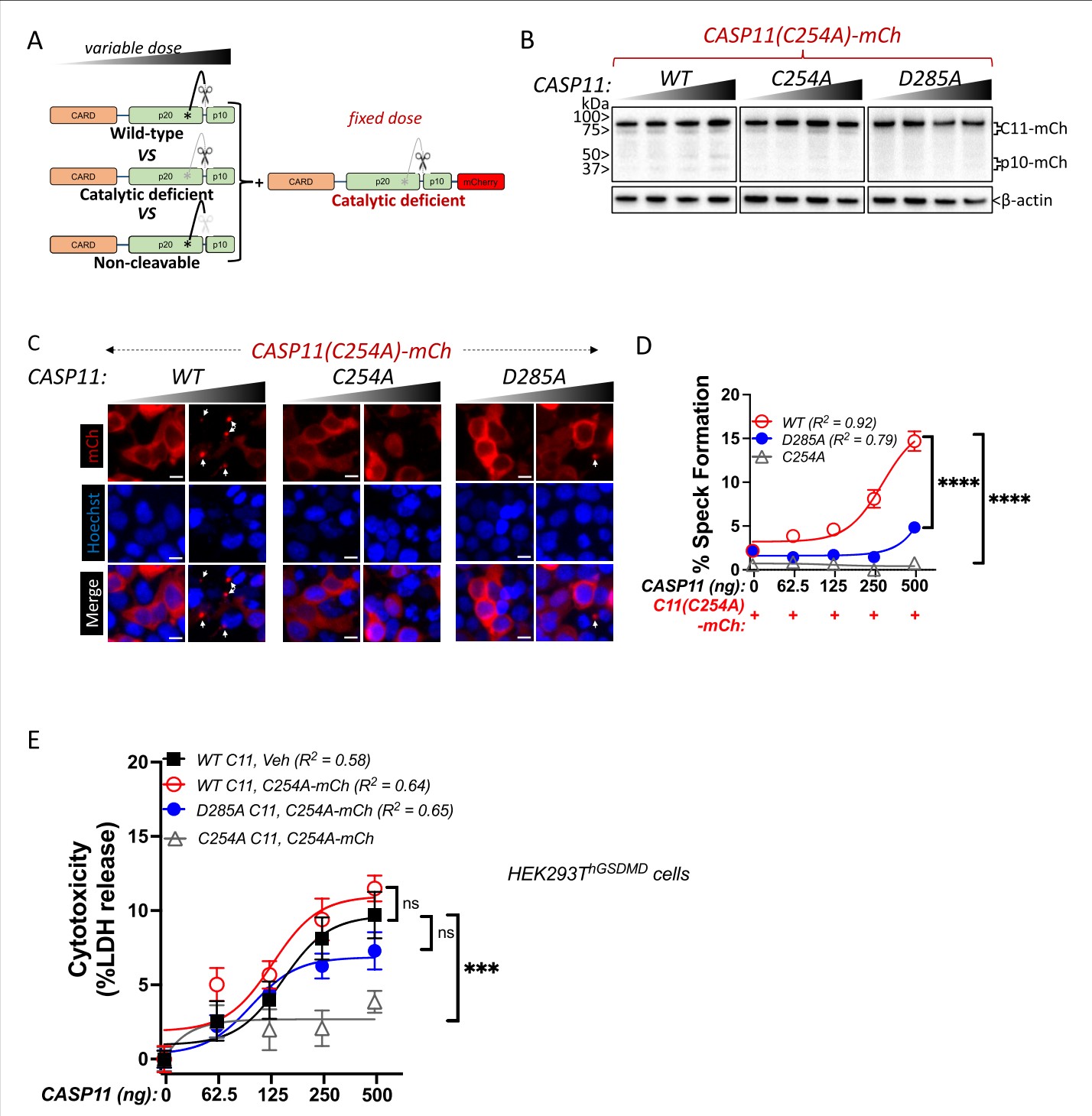

**Figure 5.** Caspase-11 autoprocessing mediates noncanonical inflammasome assembly. (**A**) Schematic diagram indicating co-transfection combinations used in (**B–D**). Unlabeled wild-type (WT), catalytically inactive (C254A), or non-cleavable (D285A) caspase-11 constructs were transfected into HEK293T cells at increasing doses, together with a fixed dose of catalytically inactive (C254A) mCherry-tagged caspase-11. (**B**) 12 hr post-transfection, whole-cell lysates were harvested and immunoblotted for mCherry or β-actin as loading control. (**C**) 18 hr following transfection, cells were imaged by fluorescence microscopy. Nuclei (blue) are stained with Hoechst, white arrows denote caspase-11 specks, scale bar = 10 μm. (**D**) Speck formation was quantified as percentage of mCherry-expressing cells containing at least one speck. (**E**) HEK293T cells stably expressing human gasdermin D (hGSDMD) were transiently transfected with a fixed dose of empty plasmid (Veh) OR mCherry-tagged C254A caspase-11, plus increasing doses of unlabeled WT, C254A, or D285A caspase-11 constructs. Cytotoxicity was measured as percent lactate dehydrogenase (LDH) release 18 hr post-plasmid transfection. All error

*Figure 5 continued on next page*

*Figure 5 continued*

bars = mean ± SEM of triplicate wells (800–900 cells per well); representative of three independent experiments. Dose–response curves were plotted by least-squares nonlinear regression ([Log$_2$(agonist) vs. response (three parameters)]; Y = Bottom + (Top-Bottom)/(1 + 10$^{(LogEC50-X)}$); R$^2$ indicated). Two-way ANOVA (highest doses) with Sidak's multiple-comparison test, ns, not significant, ***p<0.001, ****p<0.0001.

The online version of this article includes the following source data for figure 5:

**Source data 1.** Source data for *Figure 5B*.

**Source data 2.** Source data for *Figure 5C*.

**Source data 3.** Source data for *Figure 5D*.

**Source data 4.** Source data for *Figure 5E*.

required catalytic activity, consistent with our finding that catalytic activity is essential for Casp11 speck formation in response to cytosolic LPS (*Figure 6B and C*). Importantly, DmrB-(ΔCARD)-Casp11-mCherry-induced GSDMD cleavage required dimerization and catalytic activity, indicating that these constructs are functional and consistent with previous observations that dimerization induces catalytic activity of initiator and inflammatory caspases (*Figure 6D*, *Figure 6—figure supplement 1*; *Ross et al., 2018*). Our observation that CARD-deficient dimerizable Casp11 had some capacity to form specks raised the question of whether low-level spontaneous dimerization of the DmrB domain was leading to subsequent higher-order speck formation or whether the CARD itself was in fact dispensable for speck formation in this setting. Surprisingly, CARD-less Casp11-mCherry spontaneously assembled into specks in the transient 293T transfection system, and this was abrogated by mutation of the catalytic cysteine, further supporting the model that higher-order oligomerization of Casp11 in response to LPS requires its enzymatic activity (*Figure 6—figure supplement 2A, B*). Moreover, a CARD-only-citrine fusion protein was unable to form specks, in contrast to the corresponding full-length Casp11-citrine fusion, further demonstrating that the enzymatic domain is vital for Casp11 oligomerization (*Figure 6—figure supplement 2C–E*). Altogether, these data indicate that higher-order oligomerization of Casp11 requires the catalytic activity and autoprocessing function of the enzymatic domain, and that the CARD is insufficient to mediate SMOC formation in response to cytosolic LPS.

## Caspase-11 processing at the interdomain linker functions together with catalytic activity to mediate higher-order caspase-11 oligomerization

Given the requirement of Casp11 autoprocessing and catalytic activity for speck formation, we next sought to dissect the contribution of Casp11 autoprocessing at the IDL region to speck assembly. We therefore replaced the endogenous Casp11 cleavage sequence ($_{278}$LPRN<u>MEAD</u>$_{285}$) with the cleavage sequence for an exogenous protease from the tobacco etch virus (TEV) to generate TEV-cleavable mCherry-tagged Casp11: Casp11-[$_{278}$ENLYFQGA$_{285}$]-mCherry (Casp11$^{TEV}$-mCherry) (*Figure 7A*; *Chavarría-Smith et al., 2016*; *Oberst et al., 2010*). We also mutated the catalytic cysteine of TEV-cleavable Casp11, generating a catalytically inactive Casp11 that cannot induce its own cleavage but can be inducibly cleaved at the IDL region via co-expression of TEV (*Figure 7A*). Co-expression of TEV protease with either WT or C254A Casp11$^{TEV}$-mCherry in HEK293T cells resulted in dose-dependent cleavage of both WT and C254A Casp11$^{TEV}$-mCherry constructs, as expected (*Figure 7B*). Critically, TEV protease induced dose-dependent speck formation of Casp11$^{TEV}$-mCherry, demonstrating that inducible TEV-dependent cleavage of Casp11 can rescue oligomerization of the autoprocessing-deficient mutant (*Figure 7C, D*). Surprisingly, TEV protease failed to rescue speck formation in the C254A Casp11$^{TEV}$-mCherry mutant, suggesting either that Casp11 catalytic activity functions at an alternate site or that catalytic residue Cys-254 plays an additional role beyond autoprocessing in inflammasome assembly (*Figure 7C, D*). Altogether, these data indicate that Casp11 autoprocessing at the IDL is necessary, but not sufficient in the absence of catalytic activity, to induce Casp11 oligomerization, suggesting that the catalytic cysteine contributes to noncanonical speck formation via additional mechanisms independent of IDL cleavage.

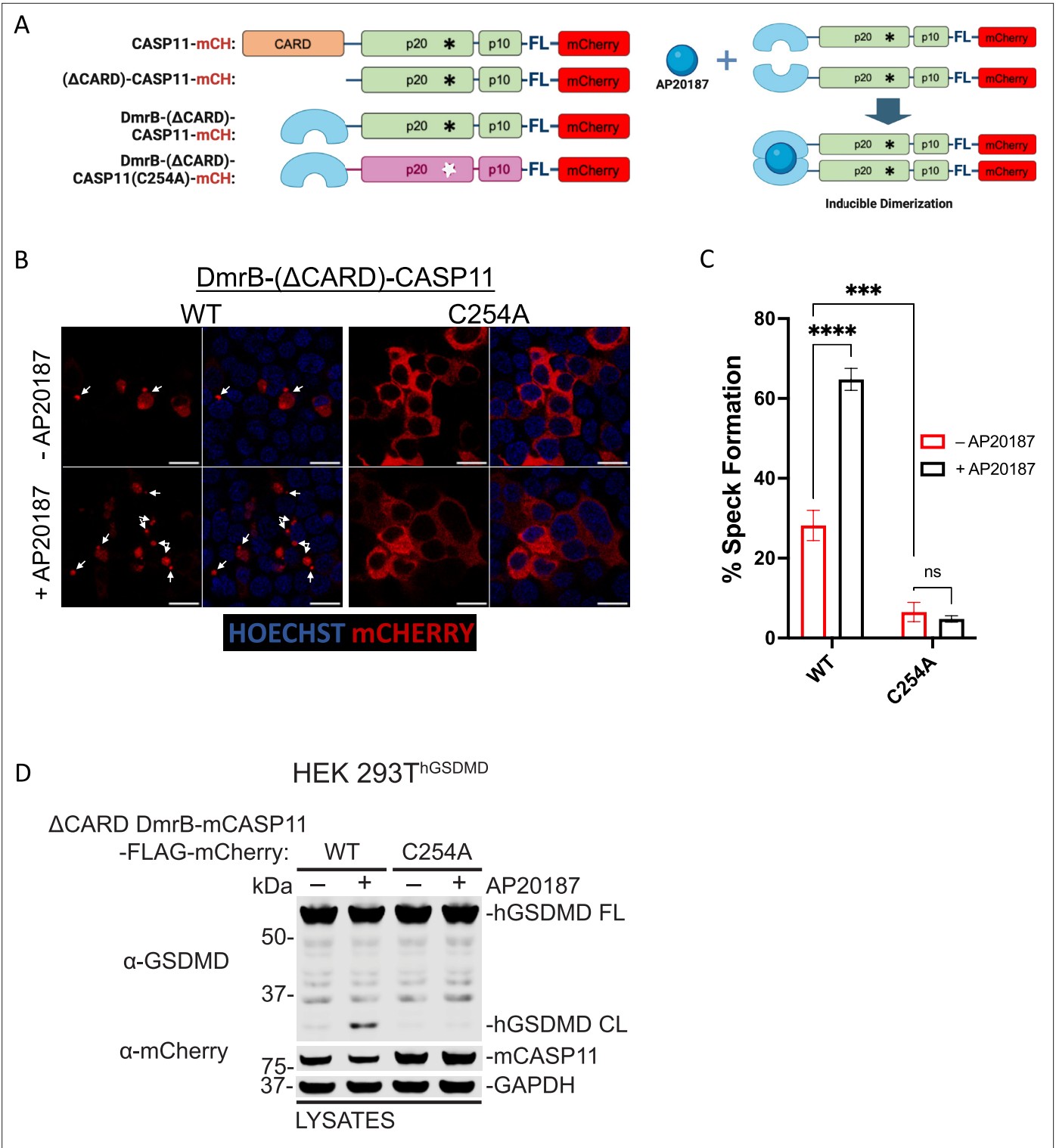

**Figure 6.** Catalytic activity is required for caspase-11 speck formation downstream of homodimerization. (**A**) Schematic representation of fluorescent Casp11 constructs that allow for inducible dimerization by the chemical dimerizer AP20187 (created with BioRender.com). (**B**) HEK293T cells were transfected with WT or catalytically inactive (C254A) DmrB-(ΔCARD)-Casp11-FLAG-mCherry constructs. 24 hr post-transfection, cells were incubated with AP20187 (1 µM) for 6 hr and imaged by confocal microscopy. Nuclei (blue) are stained with Hoechst. White arrows indicate Casp11-mCherry specks. Scale bar, 15 µm. (**C**) Speck formation in (**B**) was quantified as percentage of mCherry-expressing cells containing at least one speck. (**D**) The same constructs in (**A–C**) were transiently transfected into HEK293T cells stably expressing human gasdermin D (HEK293T$^{hGSDMD}$) and incubated in AP20187 (1 µM) for 6 hr. Lysates were immunoblotted for GSDMD and mCherry, with GAPDH as loading control. Error bars represent mean ± SEM of

*Figure 6 continued on next page*

*Figure 6 continued*

triplicate wells (800–900 cells per well); representative of three independent experiments. Data were analyzed by two-way ANOVA with Sidak's multiple-comparison test, ***p<0.001, ****p<0.0001.

The online version of this article includes the following source data and figure supplement(s) for figure 6:

**Source data 1.** Source data for *Figure 6B*.

**Source data 2.** Source data for *Figure 6D*.

**Figure supplement 1.** Inducible dimerization promotes Casp11 enzymatic activity.

**Figure supplement 1—source data 1.** Source data for *Figure 6—figure supplement 1*.

**Figure supplement 2.** Casp11 CARD is neither necessary nor sufficient to mediate spontaneous Casp11 oligomerization in HEK293T cells.

**Figure supplement 2—source data 1.** Source data for *Figure 6—figure supplement 2A*.

**Figure supplement 2—source data 2.** Source data for *Figure 6—figure supplement 2D*.

## Discussion

Casp11 is a direct sensor of cytosolic LPS and plays a vital role in host defense against infection (*Hagar et al., 2013*). Dysregulation of Casp11 contributes to severe inflammatory disorders such as endotoxic shock and experimental autoimmune encephalomyelitis (*Hagar et al., 2013*; *Kajiwara et al., 2014*; *Kayagaki et al., 2013*; *Napier et al., 2016*). LPS binds to the CARD of Casp11 and promotes its assembly into a large oligomeric complex termed the noncanonical inflammasome (*Shi et al., 2014*). Given their critical roles in innate immune responses to disease states resulting from infection, auto-inflammatory disease, and sepsis, there is substantial interest in understanding how assembly of such higher-order oligomeric complexes, also known as SMOCs, is regulated (*Kagan et al., 2014*).

Dimerization of Casp11 is sufficient to induce its protease activity and autoprocessing, which is required for its ability to cleave GSDMD, suggesting that activation of the fully competent oligomeric complex occurs subsequent to initial clustering of Casp11 (*Ross et al., 2018*). Genetic studies further reveal that both catalytic activity at C254 and autoprocessing at D285 within the IDL region between the two enzymatic subunits are required for Casp11-mediated processing of GSDMD and inflammatory function (*Lee et al., 2018*). However, the requirement for Casp11 autoprocessing in order to cleave GSDMD, even when dimerized and catalytically active (*Ross et al., 2018*), implies that dimerization alone is not sufficient for complete activation of Casp11. This requirement for both dimerization and autoprocessing in Casp11 activation parallels human CASP8 and CASP1, which also require dimerization and autoprocessing for full activity (*Ball et al., 2020*; *Oberst et al., 2010*). These findings suggest that initial dimerization and subsequent oligomerization might be two distinctly regulated steps in noncanonical inflammasome activation. Notably, in contrast to other caspases, Casp11 acts as its own sensor and adapter for assembly of the LPS-induced noncanonical inflammasome. We therefore sought to dissect the cell biological roles of catalytic activity and autoprocessing in assembly and activation of the Casp11 inflammasome.

Our findings demonstrate that cytosolic LPS induces assembly of perinuclear Casp11 clusters in BMDMs, similar to ASC inflammasome specks (*Stutz et al., 2013*; *Tzeng et al., 2016*). Surprisingly, the catalytic cysteine and IDL cleavage site of Casp11 were required to form LPS-induced Casp11 specks in macrophages. This implies that CARD binding to LPS is insufficient to mediate Casp11 SMOC formation. Consistently, the CARD domain alone was unable to mediate speck formation in response to LPS, indicating an important role for the enzymatic domain in mediating assembly of the Casp11 inflammasome complex. Importantly, however, Casp11 catalytic activity was dispensable for speck formation when BMDMs were infected with the Gram-negative bacterial pathogen *L. pneumophila*, suggesting that additional mechanisms may exist to induce Casp11 speck formation without the need for autoprocessing in the setting of bacterial infection, potentially via the ability of bacteria serving as platforms to facilitate higher-order oligomerization. Additionally, the GBPs have been implicated in liberating LPS from Gram-negative bacteria, as well as allowing for enhanced access to bacterial LPS by Casp11 (*Finethy et al., 2015*; *Meunier et al., 2014*; *Pilla et al., 2014*). GBP recruitment to the bacterial surface in the context of infection may bypass the requirement for Casp11 catalytic activity. Further studies are needed to fully determine how bacterial infection induces formation of the Casp11 inflammasome. Nonetheless, our findings collectively highlight a key role

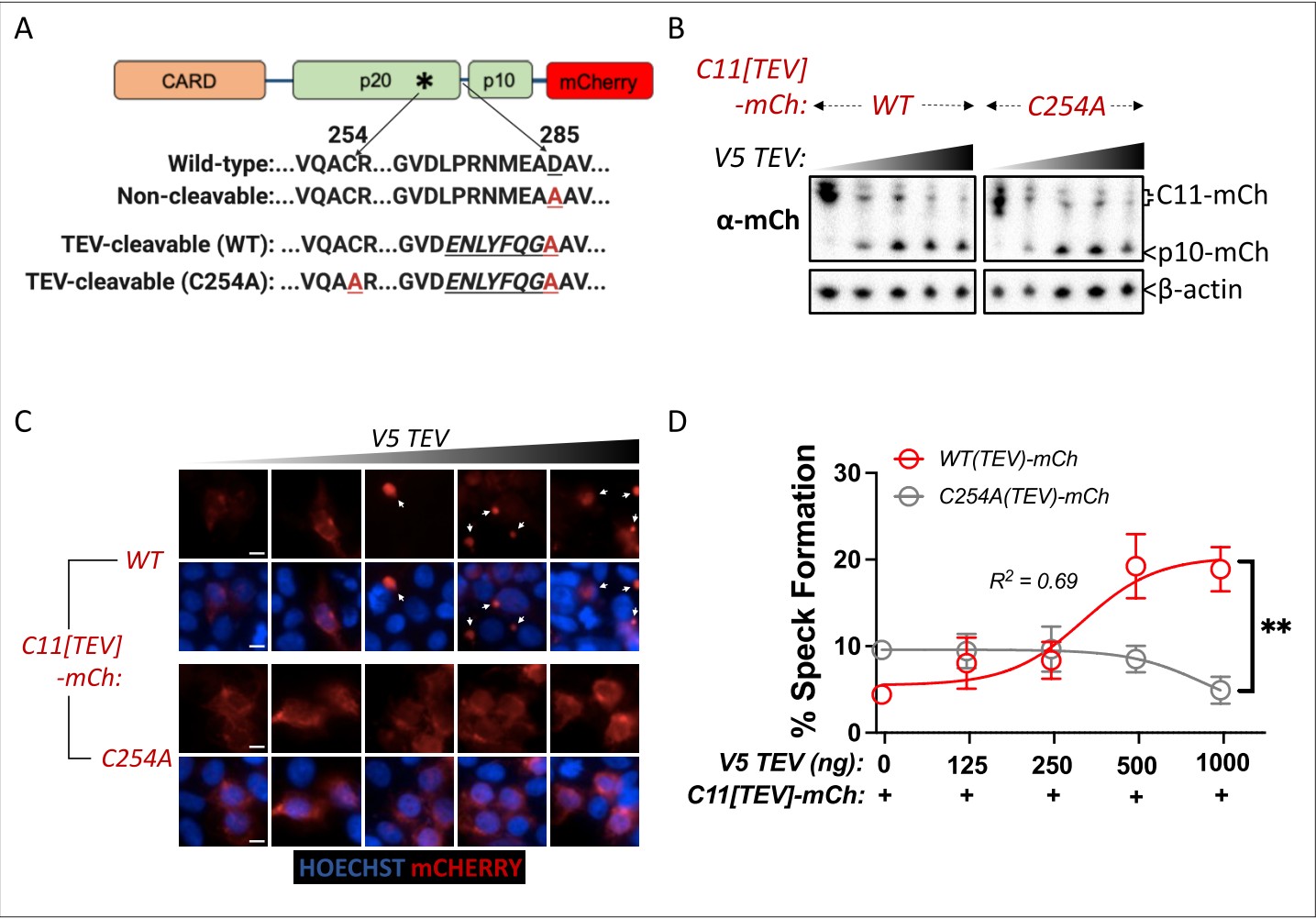

**Figure 7.** Caspase-11 processing at the interdomain linker functions together with catalytic activity to mediate higher-order caspase-11 oligomerization. (**A**) Schematic indicating Casp11-mCherry constructs that allow for inducible processing by tobacco etch virus (TEV) protease. The TEV protease consensus cleavage sequence (ENLYFQ/G) replaced the endogenous cleavage site in Casp11-mCherry constructs with preserved (WT) or mutated (C254A) catalytic activity. (**B**) HEK293T cells were transfected with the indicated TEV-cleavable Casp11-mCherry constructs, together with increasing doses of V5-TEV protease (0–500 ng). Whole-cell lysates were harvested 12 hr post-transfection and immunoblotted for mCherry or β-actin (loading control). (**C, D**) 18 hr post-transfection, cells were imaged by fluorescence microscopy and speck formation was quantified as percentage of Casp11-mCherry-expressing cells containing at least one speck. White arrows indicate Casp11-mCherry specks. Scale bar, 10 μm. Error bars represent mean ± SEM of triplicate wells (800–900 cells per well); representative of three independent experiments. Dose–response curves were plotted by least-squares nonlinear regression ([Log$_2$(agonist) vs. response (three parameters)]; Y = Bottom + (Top-Bottom)/(1 + 10$^{(LogEC50-X)}$); $R^2$ indicated). Data were analyzed by two-way ANOVA with Sidak's multiple-comparison test (highest doses), **p<0.01.

The online version of this article includes the following source data for figure 7:

**Source data 1.** Source data for *Figure 7B*.

**Source data 2.** Source data for *Figure 7D*.

for Casp11 enzymatic activity and autoprocessing in initiating assembly of a fully active noncanonical inflammasome in response to cytosolic LPS.

HEK293T cells have been used extensively to dissect the molecular and cellular determinants of inflammasome assembly and innate immune signaling (*Evavold et al., 2021*; *Kayagaki et al., 2013*; *Akaberi et al., 2020*; *Shi et al., 2014*; *Tenthorey et al., 2014*). Ectopic expression of Casp11-mCherry fusion proteins in HEK293T cells, along with corresponding catalytically inactive and autoprocessing-deficient mutants, showed that both Casp11 catalytic activity and autoprocessing are required for spontaneous recruitment of Casp11 into oligomeric complexes (*Figure 2B*). Importantly, the requirement for catalytic activity and autoprocessing in speck formation was observed in spontaneous speck

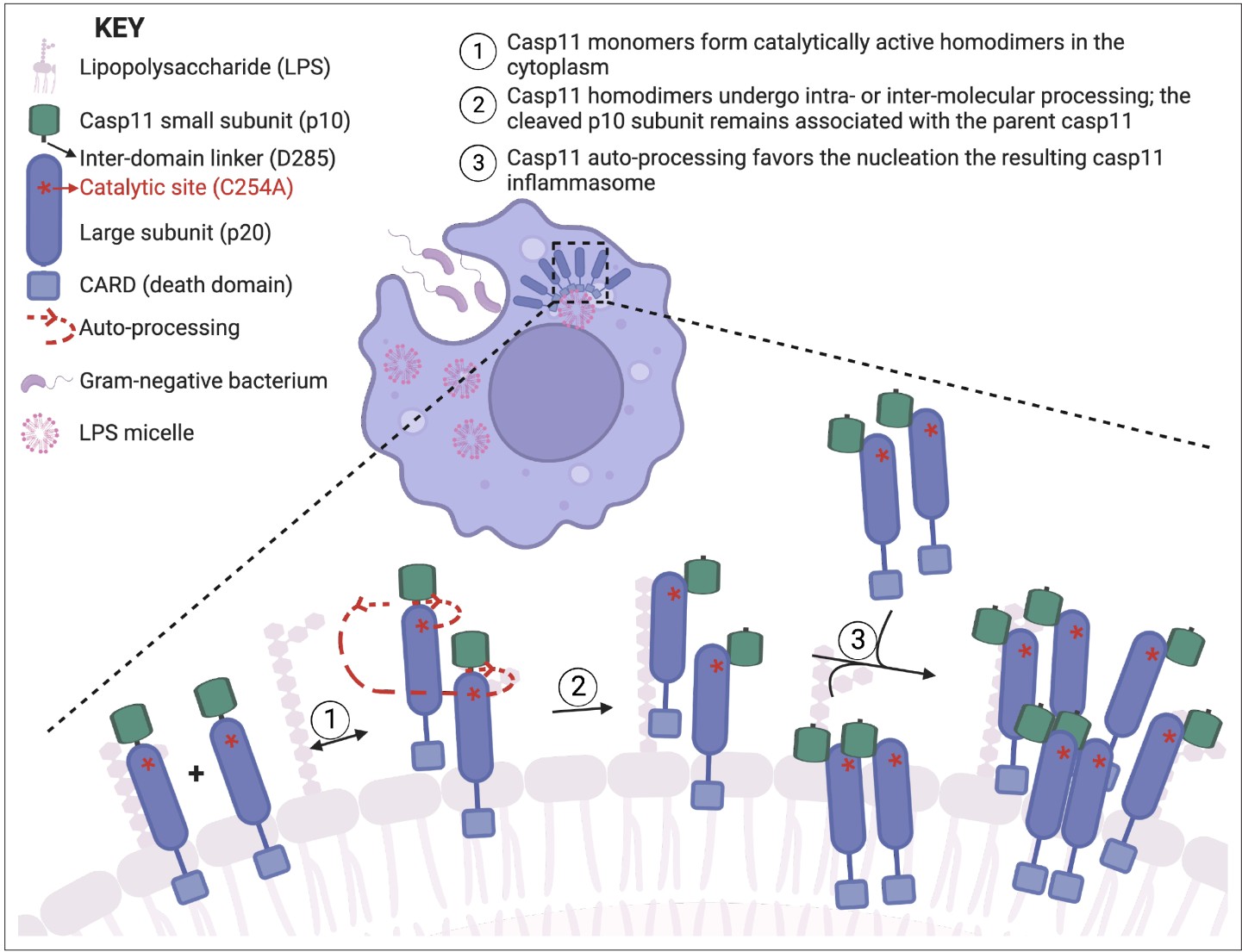

**Figure 8.** Proposed mechanism of caspase-11 inflammasome assembly (created with BioRender.com).

forming assays in HEK293T cells, in LPS-transfected primary BMDMs, and inducible speck forming assays in HEK293T cells stably expressing Casp11 (*Figures 1–3*). Collectively, these studies demonstrate that spontaneous speck assembly in HEK293T cells follows similar rules to LPS-induced speck assembly in macrophages and HEK293T cells, and that the spontaneous HEK293T system can be used to understand how the noncanonical inflammasome is assembled in response to cytosolic LPS.

Casp11 processing by the exogenous TEV protease restored speck formation of Casp11 whose endogenous IDL cleavage site was replaced with that of the TEV protease cleavage site, indicating that Casp11 processing is sufficient for speck formation (*Figure 7B–D*). Surprisingly, the Casp11 catalytic residue Cys-254 was still required for speck assembly of TEV-cleavable Casp11 (*Figure 7B–D*), suggesting that Casp11 must either undergo processing at an additional site, such as D80 of the CARD domain linker (CDL) or that the catalytic Cys-254 residue plays an alternate role in speck assembly. Notably, the CDL of Casp1 is a negative regulator of Casp1 activity, whereas the CDL of Casp11 was previously suggested to have no impact on Casp11 activity (*Boucher et al., 2018*; *Ross et al., 2018*). Thus, the alternate target of Casp11 outside of the IDL processing site is currently not clear. Importantly, mutant Casp11$^{C254A}$ was able to bind to both Casp11$^{WT}$ and Casp11$^{C254A}$, indicating that loss of binding per se was not the basis for the inability of Casp11$^{C254A}$ to undergo oligomerization (*Figure 3—source data 1*). Regardless, our studies demonstrate that in the presence of a functional catalytic

site, autoprocessing is sufficient to form the Casp11 speck, thereby implicating autoprocessing in oligomerization of noncanonical inflammasome complexes.

Our studies find that Casp11$^{WT}$ can rescue speck formation of catalytically inactive Casp11 expressed in the same cells. This occurred in both cells stably expressing Casp11$^{C254A}$ mutant protein, as well as in transient co-transfection of WT and mutant Casp11. However, exogenously expressed TEV protease was unable to perform this function despite robust processing of Casp11-mCherry, suggesting that rather than solely inducing cleavage of the catalytically inactive variant, co-expressed Casp11$^{WT}$ was recruiting inactive Casp11 into a complex formed by Casp11$^{WT}$. Indeed, Casp11$^{C254A}$-mCherry did not require the ability to be processed to form specks in the presence of Casp11$^{WT}$, whereas both catalytic activity and processing were necessary within the same molecule for Casp11 to induce speck formation of catalytically inactive Casp11$^{C254A}$-mCherry (*Figure 5*). These data suggest that intra-molecular *cis*-processing, or a combination of *cis*- and *trans*-processing, may initiate or drive assembly of higher-order Casp11 oligomeric complexes via a feed-forward mechanism whereby Casp11 dimerization and *cis*-processing, downstream of LPS binding, propagates the assembly of a fully active Casp11 SMOC. Notably, recent cryo-EM analyses revealed that human NLRP1 and CARD8 propagate their inflammasomes through a similar autoprocessing-dependent model (*Lu et al., 2021*). Likewise, cooperative assembly has been proposed for Casp8 assembly onto the Death-Inducing Signaling Complex (DISC) (*Fox et al., 2021*), suggesting that this mechanism may generally apply to other oligomeric caspase complexes.

Catalytically inactive recombinant Casp11 purified from insect cells underwent aggregation into higher-molecular-weight complexes in the presence of lysates from Gram-negative bacteria or purified LPS (*Shi et al., 2014*). These studies utilized highly purified components to define mechanics of Casp11 assembly in vitro. Here, we utilized a combination of genetic and pharmacological approaches to visualize the behavior of Casp11 within individual cells. Based on the behavior of analogous ASC specks, we interpret the Casp11 specks that we observe to be higher-order oligomeric Casp11 structures. It is possible that regulatory mechanisms within the cytosol of intact cells limit the aggregation of catalytically inactive or non-cleavable Casp11. A previous study found that EGFP-tagged Casp11 exhibited speck formation in WT and catalytically inactive Casp11-expressing 293T cells (*Liu et al., 2020*). Notably, these Casp11-EGFP specks appear smaller (<1–2 μm) and less defined than the spherical perinuclear specks (2–5 μm) we observed in Casp11-mCherry or Casp11-Citrine-expressing cells. Moreover, our observations in HEK293T cells recapitulate the endogenous behavior of the Casp11 SMOC in macrophages, as catalytic activity is required for LPS-induced Casp11 assembly in macrophages as well as spontaneous Casp11 speck assembly in 293T cells.

While guanylate-binding proteins (GBPs) facilitate Casp11 activation in response to bacterial infection (*Man et al., 2016*; *Meunier et al., 2014*; *Pilla et al., 2014*; *Wandel et al., 2020*), and in some settings, NLRP3-dependent detection of bacterial mRNA potentiates Casp11 activation (*Moretti et al., 2022*), Casp11 contains all of the properties of an inflammasome sensor, adaptor, and effector within a single protein. Interestingly, while the Casp11 CARD is sufficient to bind LPS (*Shi et al., 2014*), it is not sufficient for spontaneous Casp11 oligomerization in HEK293 cells, further supporting the finding that critical oligomerization functions are provided by catalytic activity and autoprocessing. Indeed, we found that homodimerization of CARD-less Casp11 promoted speck formation, and that this also required catalytic activity, implying that catalytic activity and autoprocessing mediate Casp11 oligomerization following initial dimerization.

Precisely how autoprocessing mediates Casp11 higher-order oligomerization remains to be determined. Initial Casp11 homodimerization and autoprocessing may induce a conformation change that enhances the binding affinity between different homodimers, resulting in a shift in the equilibrium to favor sequential recruitment of further monomers to propagate the fully assembled oligomeric complex (*Figure 8*). Collectively, our data suggest a model in which CARD-mediated LPS binding induces initial dimerization of Casp11, which then undergoes IDL processing to promote oligomerization to generate a GSDMD processing-competent complex. As both the Casp11 N- and C-termini were found in the speck, the composition of the speck may be a mixture of unprocessed, or partially processed, and fully processed Casp11 molecules. Future studies on the oligomeric structure of Casp11 are likely to reveal important information about how the individual Casp11 monomers interact with one another and how CARD-LPS interactions facilitate Casp11 oligomerization. Our findings provide new insight into the assembly of higher-order caspase complexes and reveal a role for

catalytic activity and autoprocessing upstream of noncanonical inflammasome assembly in response to cytosolic bacterial LPS.

# Materials and methods

## Cell culture

HEK293T cells were purchased from ATCC, grown in complete DMEM (supplemented with 10% v/v FBS, 10 mM HEPES, 10 mM sodium pyruvate, 1% Penicillin/Streptomycin), and maintained in a 37°C incubator with 5% $CO_2$. Murine bone marrow progenitors were harvested from femurs and hip bones of 8–12-week-old C57BL/6 or $Casp11^{-/-}$ mice and differentiated into BMDMs using 30% L929-conditioned complete DMEM for 7 d as previously described (*Bjanes et al., 2021*). Cell lines were tested to be *Mycoplasma*-free by use of the standard MycoSEQ *Mycoplasma* detection assay (Thermo Fisher Scientific).

## Cloning

The Casp11 coding sequence was fused to mCherry at its C-terminus, yielding the fusion reporter construct Casp11-mCherry encoded on the pTwist Lenti-SFFV-WPRE lentiviral vector (Twist Biosciences) or mammalian expression vector pReceiever-M56 (Genecopoeia). Caspase-11 catalytically inactive (C254A) and cleavage (D285A) mutants were generated by site-directed mutagenesis (Q5 SDM Kit, New England BioLabs; #E0554S) of the wild-type parent vector according to the manufacturer's instructions. To generate TEV-cleavable constructs, the TEV protease consensus cleavage sequence (_ENLYFQ/G_) was engineered to replace to endogenous cleavage site in WT or cleavage-deficient (D285A) Casp11-mCherry constructs on the pTwist Lenti-SFFV-WPRE lentiviral vector backbone. Catalytic site mutants (C254A) were generated as above. All constructs were confirmed by sequencing prior to experimentation.

## HEK293T transient transfections

Mammalian expression plasmids containing indicated DNA constructs were transfected into HEK293T cells using the transfection agent polyethyleneimine (PEI; Sigma-Aldrich) at 1:1 ratio (w/w DNA:PEI) in Opti-MEM (Gibco). Media was changed to complete DMEM (10% v/v FBS) after 4–6 hr, and cells were incubated for subsequent times as indicated in figure legends in a humidified incubator at 37°C and 5% $CO_2$ prior to subsequent analysis.

## Generation of stable HEK293T cell lines

To generate the HEK293T cell line ectopically expressing 2x-FLAG-Casp11 mutant, the plasmid encoding this protein was packaged into lentivirus by transfecting the vector (2 µg) along with psPAX2 (2 µg), and pMD2.G (1 µg) using Fugene HD transfection reagent (Promega) into HEK293T cells. After 2 d, the supernatants were filtered using a 0.45 µm filter, then used to infect HEK293T cells. After 48 hr, the cells expressing the indicated constructs were selected with puromycin (1 µg/mL; Sigma-Aldrich). Cell lines were tested to be *Mycoplasma*-free by use of the standard MycoSEQ *Mycoplasma* detection assay (ThermoFisher Scientific).

## Cytosolic LPS delivery, LDH cytotoxicity assay, and ELISA

Following differentiation into mature macrophages, primary murine BMDMs were seeded in TC-treated 96-well plates ($6.0 \times 10^4$ cells/well) and left to adhere in a 37°C incubator overnight. On the day of LPS transfection, cells were primed with synthetic bacterial triacylated lipopeptide Pam3CSK4 (400 ng/mL; Invivogen) for 4 hr. LPS from *Salmonella enterica* serotype Minnesota (Sigma) was then packaged into stable complexes using FuGENE HD (Promega; E2311) and added to cells in Opti-MEM (Gibco) as previously described (*Hagar et al., 2013*; *Harberts et al., 2022*). 16 hr following LPS transfection, supernatants were harvested and assayed for lactate dehydrogenase (LDH) release as a read-out for pyroptosis, as previously described (*Mariathasan et al., 2004*; *Rayamajhi et al., 2013*). Briefly, at the appropriate timepoints, plates were spun down ($250 \times g$) for 5 min to rid supernatants of cell debris. Supernatants (50 µL) were then combined with an equivalent volume of LDH reaction buffer/substrate mix (Takara Bio Inc) in a clear-bottom 96-well plate. After 20 min at room temperature (RT), absorbance was read on a spectrophotometer (495 nm) and normalized to mock-transfected cells

(minimal cell lysis; negative control) and cells treated with 1% TritonX-100 (maximal cell lysis; positive control). To assess IL-1β release, supernatants were diluted fourfold and applied to Immulon ELISA plates (ImmunoChemistry Technologies) pre-coated with anti-IL-1β capture antibody (eBioscience). Following blocking (1% BSA in 1× PBS), plates were incubated with biotin-linked secondary antibody, followed by horseradish peroxidase-conjugated streptavidin. As read-out for IL-1β levels, peroxidase enzymatic activity was determined by exposure to o-phenylenediamine hydrochloride (Sigma) in citric acid buffer. Reactions were stopped with sulfuric acid and absorbance values were read at 490 nm, normalized to mock-transfected cells (negative control).

## Lentiviral transduction and pore formation assay in BMDMs

Lentiviral plasmids encoding WT, C254A, or D285A Casp11-mCherry, along with pVSV-G and psPAX2 packaging plasmids, were transfected into HEK293T cells using PEI (Sigma-Aldrich) at 1:1 ratio (w/w DNA:PEI) in low-serum media (complete DMEM with 2% v/v FBS). Empty vector plasmids were used as negative control. After 12 hr, media was changed to complete DMEM (10% v/v FBS) and cells were incubated for another 48 hr at 37°C and 5% $CO_2$ for virus production. The resulting lentiviral supernatants were filtered (0.45 µm) and concentrated with Lenti-X concentrator (Takara Bio) according to manufacturer's instructions. Viral prep was resuspended in 30% L929-supplemented complete DMEM, supplemented with 2 µg/mL polybrene, and used to spin-infect day 2 $Casp11^{-/-}$ bone marrow progenitors (1750 RPM for 90 min at 30°C). Progenitors were replaced in 37°C incubator and allowed to mature into BMDMs for four additional days. Upon maturity, BMDMs were seeded in TC-treated 24-well plates (200,000 cells/well) and left to adhere in 37°C incubator overnight. Mature B6 BMDMs were used as positive control. Cells were then Pam3CSK4-primed and transfected with LPS as described above. Casp11-mediated pyroptosis was quantified 8 hr post-LPS transfection using a single-cell-based microscopy assay of pore formation based on uptake of Green Fluorescent (GF) Live/Dead Fixable dye (Invitrogen) according to the manufacturer's instructions. Percent pore formation was determined as percentage of Casp11-mCherry-expressing cells that stain Live/Dead positive relative to the total number of cells. Nuclei are stained with Hoechst.

## LPS transfection in stable HEK cell lines

HEK293T cells stably expressing 2x-FLAG-Casp11(WT) or 2x-FLAG-Casp11(C254A) were seeded on round, poly-L-ornithine-coated #1.5H glass coverslips (Thorlabs, #CG15NH) in 24-well plates (100,000 cells per well) and allowed to adhere overnight at 4°C. Non-transduced HEK293T cells were used as negative control. LPS (1 µg/mL) from *S. enterica* serotype Minnesota (Sigma-Aldrich) was then packaged into stable complexes using FuGENE HD (1:1 v/v; Promega E2311) and added to cells in Opti-MEM (Gibco) as described above. For cells incubated with pan-caspase inhibitor zVAD, zVAD (200 µM) was added to the appropriate wells at the time of LPS transfection. Cells were incubated for 24 hr at 37°C/5% $CO_2$. Following incubation, cells were washed 2× with PBS and fixed with freshly prepared 1:1 acetone:methanol for 5 min at RT. Cells were washed 3× with PBS and permeabilized in 0.2% TritonX in PBS at RT for 10 min. Then, cells were washed 3× with PBS and blocked with 10% BSA at RT on a shaker for 1 hr. Cells were washed 3 × 5 min with 0.2% Tween-20 and 1.5% BSA in PBS on a rocker. Cells were incubated with anti-FLAG-FITC antibody for 1 hr in the dark, and then washed 4 × 5 min with 0.2% Tween-20 and 1.5% BSA in PBS on a rocker. Coverslips were mounted on glass slides with DAPI-containing mounting media and dried overnight. For colocalization studies, 2xFLAG-Casp11-expressing HEK293Ts were seeded as described above, transfected with Casp11-mCherry plasmids using FuGENE HD, and incubated for 24 hr prior to fixation and staining as described in this section. All slides were imaged on Zeiss LSM 980 Confocal using the ×63 objective. Slides were imaged at a single z-plane per field with lasers optimized for FITC (488-green), mCherry (561-red), and DAPI (405-blue) emission spectra.

## *Legionella* intracellular infection

*L. pneumophila* serogroup 1 strains were used in all experiments. Lp02-derived (thymidine auxotroph) flagellin-deficient (*ΔflaA*) *L. pneumophila* was cultured on charcoal yeast extract (CYE) agar plates for 48 hr at 37°C prior to infection. Following differentiation into mature macrophages, primary murine BMDMs were seeded in TC-treated 96-well plates (5.0 × 10⁴ cells/well) and left to adhere in 37°C/5% $CO_2$ incubator overnight. On the day of infection, BMDMs were infected with mock or *ΔflaA*

*L. pneumophila* at a multiplicity of infection (MOI) of 50, and incubated for 6 hr in a 37°C/ 5% $CO_2$ incubator (*Casson et al., 2013*; *Ren et al., 2006*). The cells were then fixed with 4% paraformaldehyde for 10 min, washed 3× with PBS, and stained for 10 min with Hoechst (1 μg/mL). The cells were then washed 3× with PBS and stored in PBS at 4°C until imaging.

## Cell viability assay

Viability of HEK293T cells co-transfected with GSDMD and various caspase-11 expression constructs was determined using the CellTiter-Glo 2.0 Assay Kit (CTG2.0; Promega) according to the manufacturer's instructions. Briefly, HEK293T cells were seeded in poly-L-lysine-coated (0.1 mg/mL; Sigma-Aldrich) F-bottom 96-well plates and allowed to adhere overnight in complete DMEM. The next day, mammalian expression plasmids containing indicated mCherry-tagged or -untagged caspase-11 DNA constructs were co-transfected into the cells at increasing doses (0–250 ng per well), together with a fixed dose of GSDMD (50 ng), using the transfection agent polyethylenimine (PEI; Sigma-Aldrich) at a 1:1 ratio (w/w DNA:PEI). After 14 hr, cells were lysed with CTG2.0 reagent mix and incubated in the dark at 37°C for 30 min. Luminescence was read on a luminometer, and values were normalized to cells treated with 1% TritonX-100 (min cell viability; negative control) and mock-transfected cells (max cell viability).

## Immunoblotting

BMDMs were seeded in clear TC-treated 96-well plates ($6.0 \times 10^4$ cells/well) and transfected with LPS in the absence or increasing doses of zVAD-fmk (caspase-11 inhibitor), as indicated. HEK293T cells were seeded in poly-L-lysine-coated TC-treated 24-well plates ($2.0 \times 10^5$ cells/well) and transiently transfected with appropriate constructs. At the indicated timepoints, whole-cell extracts (XT) and/or supernatants (sup) were harvested and immunoblotted for various proteins as previously described (*Bjanes et al., 2021*; *Harberts et al., 2022*). In brief, plates were centrifuged ($250 \times g$) to rid supernatants of cell debris. Proteins in supernatants were then precipitated by incubating with 0.61 N trichloroacetic acid (Sigma-Aldrich) plus 1× protease inhibitor cocktail (PIC; Sigma-Aldrich) for at least 1 hr on ice. Precipitates were washed with acetone by centrifugation (×3) at 4°C and resuspended in protein sample buffer (125 mM Tris, 10% SDS, 50% glycerol, 0.06% bromophenol blue, 1% β-mercaptoethanol, 50 mM dithiothreitol). Proteins from combined supernatant and lysate samples were precipitated using methanol and chloroform. Lysates were harvested in lysis buffer (20 mM HEPES, 150 mM NaCl, 10% glycerol, 1% TritonX-100, 1 mM EDTA, pH 7.5) supplemented with 1×PIC and 1×protein sample buffer and rocked gently for 10 min at 4°C. Protein levels were normalized using the DC Protein Assay Kit (Bio-Rad). Protein samples from lysates and/or supernatants were then prepared for SDS-PAGE by boiling and centrifugation (tabletop max speed; 5 min) before they were run on 4–12% polyacrylamide gels (Invitrogen). Proteins were transferred to polyvinylidene difluoride (PVDF) or nitrocellulose membranes (Bio-Rad) and immunoblotted with the following antibodies: caspase-11 rat (1:1000; Novus Biologicals 17D9), mCherry rabbit (1:1000; Abcam EPR20579), GSDMD rabbit (1:500; Abcam EPR19828), FLAG (1:1000; Cell Signaling 2368S). β-actin mouse monoclonal Ab (1:2500; Sigma-Aldrich AC74), or GAPDH rabbit monoclonal Ab (1:2500; Cell Signaling Tech 14C10). Blots were then incubated in species-specific, horseradish peroxidase-conjugated secondary antibodies (1:2500) and imaged on the Odyssey M Imaging System (LI-COR Biosciences) or by chemiluminescence using Pierce SuperSignal West Femto maximum sensitivity substrate (ThermoFisher #34095).

## Casp11 immunoprecipitation

HEK293T cells were transiently transfected with the indicated FLAG-tagged construct for 48 hr. Collected cell pellet samples were lysed by sonication and clarified by centrifugation at $20,000 \times g$ for 5 min at 4°C. Aliquots from the clarified lysate samples were incubated with ANTI-FLAG M2 affinity beads (MilliporeSigma) in Pierce Micro-Spin columns (Thermo Fisher Scientific) at 4°C overnight. The Micro-Spin samples were then washed three times with one-column volume of PBS, and proteins eluted with 3x-FLAG peptide followed by western blot analysis. Aliquots of the remaining lysate were used to prepare input samples for western analysis.

## Fluorescence and confocal microscopy

BMDMs or HEK293T cells were seeded on round, poly-L-lysine-coated #1.5H glass coverslips (Thorlabs, #CG15NH) and allowed to adhere overnight. Cells were then transfected with LPS (BMDMs) or indicated DNA constructs (HEK293Ts). At the indicated timepoints, cells were washed 1× with PBS and fixed with 4% paraformaldehyde. Following nuclear counterstain with Hoechst 33342 (1 µg/mL; Thermo Fisher #62249), cells were mounted on glass slides with Fluoromount-G (SouthernBiotech; 0100-01) and dried overnight. For wide-field fluorescence microscopy, slides were imaged on brightfield, Cy3 (red) and DAPI (blue) fluorescence channels using a Dmi8 inverted wide-field apparatus at ×20 objective (Leica Biosystems). For confocal microscopy, slides were imaged at a single z-plane per field with lasers optimized for Cy5 (far-red), citrine (yellow), and CellTracker Biolet (Blue) emission spectra, through ×63 objective.

## Image quantification and analysis

Each experiment was conducted in three technical replicates. Within each replicate, 4–6 frames were analyzed for an average of 100–150 cells (BMDMs) or 800–900 cells (HEK293T cells) per well. A speck was defined as a distinct high-fluorescent perinuclear cluster of mCherry, anti-FLAG-FITC, or citrine. Speck formation frequency was determined as the percentage of mCherry- (red), FLAG- or citrine-expressing (yellow) cells that contained one or more specks, using custom macros from ImageJ (NIH) and LAS X (Leica Biosystems).

## Statistical analysis

Data were graphed and analyzed using GraphPad Prism 9 (San Diego, CA). Mean values (± SEM) were compared across triplicate conditions, and p-values were determined using one-sample $t$-test, one-way or two-way ANOVA with Sidak's multiple comparison. Dose–response curves were plotted by least squares nonlinear regression curves on GraphPad Prism 9. For agonists: [$\log_2$(agonist) vs. response (three parameters)]; $Y = Bottom + (Top-Bottom)/(1 + 10^{(LogEC50-X)})$. For inhibitors, [$\log_2$(inhibitor) vs. response (three parameters)]; $Y = Bottom + (Top-Bottom)/(1 + 10^{(X-LogIC50)})$. All $R^2$ values are indicated where applicable.

## Acknowledgements

We are grateful to members of the Brodsky, Shin, and Taabazuing labs for scientific discussion. We would like to thank the Herbert and Vaughan labs for use of lab equipment and the Leica Dmi8 inverted wide-field microscope. We would also like to thank the Cell & Developmental Biology Microscopy Core at the Perelman School of Medicine at the University of Pennsylvania for access and usage of the Zeiss LSM 980 microscope. This work was supported by grants R01AI128530, R01139102A1, and awards from the Mark Foundation (IEB), American Heart Association Predoctoral Fellowship 916272 (DCA), K12GM081259 (KAW), F31AI172200 (RSW) Martin and Pamela Winter Infectious Disease Fellowship (PME), National Science Foundation Predoctoral Fellowship DGE-1650114 (VRVM), R01AI118861 and R01AI123243 (SS), the Burroughs Wellcome Fund Investigator in the Pathogenesis of Infectious Disease Award (IEB and SS), and UNCF/BMS EE Just Early Career Investigator Award and NIH R00 Career Transition Award Grant #4R00AI148598-03 (CYT).

## Additional information

### Funding

| Funder | Grant reference number | Author |
| --- | --- | --- |
| National Institute of Allergy and Infectious Diseases | R01AI128530 | Igor E Brodsky |
| National Institute of Allergy and Infectious Diseases | R01139102A1 | Igor E Brodsky |
| Mark Foundation For Cancer Research | | Igor E Brodsky |

| Funder | Grant reference number | Author |
|---|---|---|
| American Heart Association | Predoctoral Fellowship, 916272 | Daniel C Akuma |
| Martin and Pamela Winter Infectious Disease Fellowship | | Patrick M Exconde |
| National Institute of Allergy and Infectious Diseases | R01AI118861 | Sunny Shin |
| National Institute of Allergy and Infectious Diseases | R01AI123243 | Sunny Shin |
| Burroughs Wellcome Fund | Investigator in the Pathogenesis of Infectious Disease Award | Sunny Shin |
| United Negro College Fund | BMS Ernest E. Just Early Career Investigator Award | Cornelius Taabazuing |
| National Institute of Allergy and Infectious Diseases | 4R00AI148598-03 | Cornelius Taabazuing |
| National Institute of General Medical Sciences | K12GM081259 | Kimberly A Wodzanowski |
| National Science Foundation | Graduate Research Fellowship DGE-1650114 | Víctor R Vázquez Marrero |
| National Institute of Allergy and Infectious Diseases | F31AI172200 | Ronit Schwartz Wertman |

The funders had no role in study design, data collection and interpretation, or the decision to submit the work for publication.

## Author contributions

Daniel C Akuma, Conceptualization, Data curation, Formal analysis, Investigation, Visualization, Methodology, Writing – original draft, Writing – review and editing; Kimberly A Wodzanowski, Ronit Schwartz Wertman, Conceptualization, Data curation, Formal analysis, Investigation, Visualization, Methodology, Writing – review and editing; Patrick M Exconde, Data curation, Formal analysis, Investigation, Writing – review and editing; Víctor R Vázquez Marrero, Methodology, Writing – review and editing; Chukwuma E Odunze, Formal analysis; Daniel Grubaugh, Conceptualization, Investigation, Writing – review and editing; Sunny Shin, Conceptualization, Supervision, Writing – review and editing; Cornelius Taabazuing, Conceptualization, Supervision, Investigation, Writing – review and editing; Igor E Brodsky, Conceptualization, Supervision, Funding acquisition, Project administration, Writing – review and editing

## Author ORCIDs

Daniel C Akuma http://orcid.org/0000-0003-1079-6295
Kimberly A Wodzanowski http://orcid.org/0000-0001-8948-6788
Ronit Schwartz Wertman http://orcid.org/0000-0003-4297-3495
Sunny Shin http://orcid.org/0000-0001-5214-9577
Cornelius Taabazuing http://orcid.org/0000-0003-2361-5457
Igor E Brodsky https://orcid.org/0000-0001-7970-872X

## Ethics

All animals were handled in compliance with the federal regulations set forth in the Animal Welfare Act (AWA), the recommendations in the Guide for the Care and Use of Laboratory Animals of the National Institutes of Health, and the guidelines of the University of Pennsylvania Institutional Animal Use and Care Committee (IACUC). All protocols used in this study were approved by the IACUC at the University of Pennsylvania (AWA Protocol #804523).

## Decision letter and Author response

Decision letter https://doi.org/10.7554/eLife.83725.sa1
Author response https://doi.org/10.7554/eLife.83725.sa2

## Additional files

### Supplementary files
• Supplementary file 1. Tables of plasmids, cell lines and oligonucleotides used in this study.

• Transparent reporting form

### Data availability
All data generated or analyzed during the study are included in the manuscript and supporting files. Source Data files have been included for *Figures 1–7*, *Figure 2—figure supplements 1–3*, *Figure 3—figure supplement 1*, *Figure 4—figure supplement 1*, *Figure 6—figure supplements 1 and 2*. All materials and reagents have been listed in the 'Materials and methods' section and *Supplementary file 1*. Studies were designed and data were analyzed and reported according to ARRIVE guidelines (*Kilkenny et al., 2010*).

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
