## [Editor Report]

This fundamental work advances our understanding of how caspase-11 is regulated by LPS. The evidence supporting the conclusions is compelling, with rigorous biochemical and cellular studies. The work will be of broad interest to immunologists and biochemists.

---

## [Decision Letter]

**Decision letter after peer review:**

Thank you for submitting your article "Noncanonical inflammasome assembly requires caspase-11 catalytic activity and intra-molecular autoprocessing" for consideration by *eLife*. Your article has been reviewed by 3 peer reviewers, one of whom is a member of our Board of Reviewing Editors, and the evaluation has been overseen by Carla Rothlin as the Senior Editor. The reviewers have opted to remain anonymous.

Essential revisions:

Please address the following issues:

1) Whether the location of the tag influences the oligomerization activity of casp11.

2) Whether intracellular pathogens yield the same results as seen from LPS.

3) Further explore and address concerns regarding the results presented in Figures 4/5 (trans/cis cleave).

Please also respond to all other concerns raised by the reviewers.

*Reviewer #1 (Recommendations for the authors):*

This is a very well-designed and timely study, nevertheless, there are a few aspects that can be strengthened.

In Figure 1, cytotoxicity induction for a transduced system is pretty low. Besides LDH measurement, it would be important to measure cell permeability and/or stain for live/dead cells (zombie dyes, etc). Also, the use of glycine to prove the cells are encountering pyroptosis can be used as an important control. A more defined analysis of pyroptosis may also be useful for the suggestions made for Figures 4 and 5.

In Figure 2 (and related supplementary figures), since the authors used a pan-caspase inhibitor, it would be important to show that death is driven by pyroptosis (either by using glycine, and/or HEK cells not transfected with GSDMD).

Data shown in Figure 3D, F are very interesting but challenge the model proposed. It is important to better highlight, both in the abstract and the introduction, that cleavage of casp11 itself is not sufficient to induce the oligomerization/speck formation and that additional roles are played by Cys-254. Notably, Figures 3A and B show a slight reduction in the penetrance of the phenotype compared to the enzymatic mutant. This is more clearly seen in Supp. Figure 1D shows a 50% reduction in cell death. Do the roles of Cys-254 partially overcome the mutation in the cleavage site? The authors may want to discuss this evidence together.

Figure 4 lacks the analyses of the level of pyroptosis, which should be not paired with the presence of mCherry specks when C254A-expressing cells also express wt Casp11. Data in Figure 4G, H further support a model in which single or double mutant casp11 insert in specks that are formed by the WT casp11. Are these specks less or more functional in terms of pyroptosis? Do the catalytic inactive and/or cleavage site mutant casp11 compete with unlabeled wt casp11 for GSDMD cleavage? Also, and in contrast to mutant casp11, is the increase in WTmCh Speck formation at the highest dose of WT casp11 in figure 4D associated with increased apoptosis?

Similar to figure 4, to strengthen the conclusions of the authors in Figure 5, pyroptosis should be measured to assess the contribution of enzymatically inactive and/or non-cleavable caspases that reach the non-canonical inflammasome as a consequence of the activity of non-labelled wt casp11.

*Reviewer #2 (Recommendations for the authors):*

This is an elegant study, the manuscript and figures are clear, and conclusions are supported by data.

One experiment would be to try to biochemically crosslink and isolate endogenous, untagged Casp11 specks upon LPS transfection of primed macrophages (e.g. after priming through IFNs or TLRs). This would mimic the natural upregulation and activation of endogenous Casp11.

Another question is whether this model applies to human orthologues, Casp4/5.

And finally, what happens after actual intracellular pathogen detection when the pathogen itself serves as a signalling platform? Are specks stills formed (or even needed)?

---

## [Author Response]

Essential revisions:Please address the following issues:1) Whether the location of the tag influences the oligomerization activity of casp11.

We thank the reviewers for this suggestion. To address this question, we have utilized a stably-expressing N-terminal FLAG-tagged CASP11 construct in HEK cells. Critically, the N-terminal FLAG CASP11-expressiing HEK cells also display speck formation that is not observed in the presence of the caspase inhibitor zVAD or with the catalytic mutant C254A CASP11, matching our BMDM data as well as the data observed with the C-terminal CASP11-mCherry construct. Importantly, in the stable expression system, we do not observe spontaneous speck formation, but rather, observe LPS-inducible speck formation, consistent with our findings in BMDMs. Altogether, these data conclusively demonstrate that the location of the tag does not influence the behavior of CASP11 specks, and that catalytic activity is important for LPS-induced speck formation regardless of the tag location. These data are now found in new Figure 3 and discussed on pp. 6-7.

2) Whether intracellular pathogens yield the same results as seen from LPS.

We very much appreciate this question, and also believe it is important. We therefore tested whether the intracellular pathogen *Legionella pneumophila* also induces Casp11 specks (new Figure 1F and G). Indeed, consistent with our initial observations, *Legionella* infection induces Casp11 specks in BMDMs expressing Casp11-mCherry. However, unlike cytosolic LPS, we observed that *Legionella*-induced speck formation is independent of catalytic activity, suggesting that whole organisms might initiate and propagate the noncanonical inflammasome through additional mechanisms beyond catalytic activity and autoprocessing. These data are now included in Figure 1 and discussed further in the results (p. 5) and discussion (p. 11) sections of the revised manuscript.

3) Further explore and address concerns regarding the results presented in Figures 4/5 (trans/cis cleave).

We agree that the data do not definitively demonstrate that *‘cis’* cleavage is occurring exclusively. Our data do demonstrate that the same molecule needs to be catalytically active and competent for processing in order to form specks and to recruit a catalytic mutant to the speck. We have therefore modified our discussion of these data to focus on autoprocessing generally, and raise the possibility of *‘cis’* cleavage as being important in speck formation in the discussion.

Reviewer #1 (Recommendations for the authors):This is a very well-designed and timely study, nevertheless, there are a few aspects that can be strengthened.In Figure 1, cytotoxicity induction for a transduced system is pretty low. Besides LDH measurement, it would be important to measure cell permeability and/or stain for live/dead cells (zombie dyes, etc). Also, the use of glycine to prove the cells are encountering pyroptosis can be used as an important control. A more defined analysis of pyroptosis may also be useful for the suggestions made for Figures 4 and 5.

We are grateful for this suggestion. The relatively low cytotoxicity is due to the low transduction efficiency of primary BMDMs, as the overall number of transduced primary BMDMs is quite low, thereby leading to low cell death signal in a bulk assay such as LDH. Per the reviewer’s suggestion, we have instead utilized a single cell-based membrane integrity/pore formation assay based on live/dead zombie dye uptake. The new data are included in new Figure 1B of the revised manuscript.

In Figure 2 (and related supplementary figures), since the authors used a pan-caspase inhibitor, it would be important to show that death is driven by pyroptosis (either by using glycine, and/or HEK cells not transfected with GSDMD).

We are grateful to the reviewer for this insightful suggestion. In Figure 2—figure supplement 1, which describes experiments involving co-transfection of Casp11-mCherry with GSDMD constructs into HEK293T cells, the cytotoxicity (Figure 2—figure supplement 1B) and cell viability (Figure 2—figure supplement 1C) were both normalized to similarly treated 293T cells lacking GSDMD. Thus, any elevated cell death can robustly be attributed to casp11-mediated pyroptosis. Studies in Figure 2D-F involve transfection of HEK293T cells with Casp11-mCherry alone; as HEK293T cells lack robust constitutive levels of other pyroptosis machinery, these cells do not undergo pyroptosis. Nonetheless, zVAD demonstrated a dose-dependent inhibition of Casp11 self-cleavage (as shown by diminishing levels of p10-mCherry in Figure 2D). Thus, although zVAD is a pan-caspase inhibitor, we concluded that its impact on CASP11-mCherry speck formation (Figure 2E, F) is due to Casp11 antagonism.

Data shown in Figure 3D, F are very interesting but challenge the model proposed. It is important to better highlight, both in the abstract and the introduction, that cleavage of casp11 itself is not sufficient to induce the oligomerization/speck formation and that additional roles are played by Cys-254. Notably, Figures 3A and B show a slight reduction in the penetrance of the phenotype compared to the enzymatic mutant. This is more clearly seen in Supp. Figure 1D shows a 50% reduction in cell death. Do the roles of Cys-254 partially overcome the mutation in the cleavage site? The authors may want to discuss this evidence together.

We appreciate this point and agree that the autoprocessing mutant does not have quite as strong a phenotype in downstream outcomes (LDH release or live/dead dye uptake) as it has on speck formation. This is consistent with findings in the literature for other caspases, such as caspase-8 and caspase-1. We have updated the manuscript to reflect this point.

Figure 4 lacks the analyses of the level of pyroptosis, which should be not paired with the presence of mCherry specks when C254A-expressing cells also express wt Casp11. Data in Figure 4G, H further support a model in which single or double mutant casp11 insert in specks that are formed by the WT casp11. Are these specks less or more functional in terms of pyroptosis? Do the catalytic inactive and/or cleavage site mutant casp11 compete with unlabeled wt casp11 for GSDMD cleavage? Also, and in contrast to mutant casp11, is the increase in WTmCh Speck formation at the highest dose of WT casp11 in figure 4D associated with increased apoptosis?Similar to figure 4, to strengthen the conclusions of the authors in Figure 5, pyroptosis should be measured to assess the contribution of enzymatically inactive and/or non-cleavable caspases that reach the non-canonical inflammasome as a consequence of the activity of non-labelled wt casp11.

We appreciate these reviewer suggestions. We have included two new figures to test whether pyroptosis and GSDMD cleavage are impacted in cells co-expressing C254A-mCherry and unlabeled WT Casp11 (Figure 4—figure supplement 1A-C and Figure 5E). Importantly, activity of WT Casp11 is neither enhanced nor inhibited by co-transfection with catalytically inactive or noncleavable caspase-11 mutants.

Reviewer #2 (Recommendations for the authors):This is an elegant study, the manuscript and figures are clear, and conclusions are supported by data.One experiment would be to try to biochemically crosslink and isolate endogenous, untagged Casp11 specks upon LPS transfection of primed macrophages (e.g. after priming through IFNs or TLRs). This would mimic the natural upregulation and activation of endogenous Casp11.

We thank the reviewer for this excellent suggestion. We are in the process of establishing systems to isolate and biochemically characterize endogenous Casp11 oligomers. Nonetheless, we believe that more extensive biochemical analysis of Casp11 oligomers is outside the scope of this current work, whose focus is on the cell biology of Casp11 oligomerization in response to cytosolic LPS. We believe that our new experiments to address reviewers’ concerns provide new insight and strong support for the catalytic activity model. Our future studies will involve more biochemical analysis of endogenous Casp11 complexes.

Another question is whether this model applies to human orthologues, Casp4/5.

This is indeed an interesting and important question. Whether the human orthologs also behave this way, and whether other initiator or inflammatory caspases also behave this way are important questions. However, here too we believe that experimentally addressing this question will involve extensive further analysis that is beyond the scope of the current work, and is being pursued in ongoing studies in our labs.

And finally, what happens after actual intracellular pathogen detection when the pathogen itself serves as a signalling platform? Are specks stills formed (or even needed)?

We thank the reviewer for this important question, which we agree is important to address here. We have tested whether an intracellular pathogen can induce specks (Figure 1F-G). Like purified LPS, non-flagellated *Legionella pneumophila* induces Casp11 specks in BMDMs. Interestingly, catalytic mutant Casp11-mCherry also formed specks robustly in the presence of intracellular *L. pneumophila*, suggesting that other structural components of the whole bacteria might dictate Casp11 oligomerization in a manner that supersedes Casp11 catalytic activity. This is discussed in the new Results section for Figure 1 (p. 5), as well as in the Discussion section (p. 11).